# Discrete symmetries tested at $10^{-4}$ precision using linear polarization of photons from positronium annihilations

Paweł Moskal [1,2], Eryk Czerwiński [1,2] ✉, Juhi Raj [1,2], Steven D. Bass [2,3], Ermias Y. Beyene [1,2], Neha Chug [1,2], Aurélien Coussat [1,2], Catalina Curceanu [4], Meysam Dadgar [1,2], Manish Das [1,2], Kamil Dulski [1,2], Aleksander Gajos [1,2], Marek Gorgol [5], Beatrix C. Hiesmayr [6], Bożena Jasińska [5], Krzysztof Kacprzak [1,2], Tevfik Kaplanoglu [1,2], Łukasz Kapłon [1,2], Konrad Klimaszewski [7], Paweł Konieczka [7], Grzegorz Korcyl [2,8], Tomasz Kozik [1,2], Wojciech Krzemień [9], Deepak Kumar [1,2], Simbarashe Moyo [1,2], Wiktor Mryka [1,2], Szymon Niedźwiecki [1,2], Szymon Parzych [1,2], Elena Pérez del Río [1,2], Lech Raczyński [7], Sushil Sharma [1,2], Shivani Choudhary [1,2], Roman Y. Shopa [7], Michał Silarski [1,2], Magdalena Skurzok [1,2], Ewa Ł. Stępień [1,2], Pooja Tanty [1,2], Faranak Tayefi Ardebili [1,2], Keyvan Tayefi Ardebili [1,2], Kavya Valsan Eliyan [1,2] & Wojciech Wiślicki [7]

Discrete symmetries play an important role in particle physics with violation of CP connected to the matter-antimatter imbalance in the Universe. We report the most precise test of P, T and CP invariance in decays of ortho-positronium, performed with methodology involving polarization of photons from these decays. Positronium, the simplest bound state of an electron and positron, is of recent interest with discrepancies reported between measured hyperfine energy structure and theory at the level of $10^{-4}$ signaling a need for better understanding of the positronium system at this level. We test discrete symmetries using photon polarizations determined via Compton scattering in the dedicated J-PET tomograph on an event-by-event basis and without the need to control the spin of the positronium with an external magnetic field, in contrast to previous experiments. Our result is consistent with QED expectations at the level of 0.0007 and one standard deviation.

Positronium, Ps, is a bound state of an electron and positron with its physics governed by quantum electrodynamics, QED. For describing Ps one commonly uses non-relativistic QED bound state theory. While this approach is mostly successful, recent hyperfine structure, HFS, spectroscopy measurements have revealed a 4.5 standard deviations anomaly between experiment and theory at the level of one part in $10^4$ (see ref. 1) prompting new thinking about Ps structure and interactions —for recent discussion see refs. 2,3.

Here, we investigate the properties of the ortho-positronium, o-Ps, spin with respect to discrete symmetries. As a bound state, o-Ps

[1]Marian Smoluchowski Institute of Physics, Jagiellonian University, Kraków, Poland. [2]Centre for Theranostics, Jagiellonian University, Kraków, Poland. [3]Kitzbühel Centre for Physics, Kitzbühel, Austria. [4]INFN, Laboratori Nazionali di Frascati, Frascati, Italy. [5]Institute of Physics, Maria Curie-Skłodowska University, Lublin, Poland. [6]Faculty of Physics, University of Vienna, Vienna, Austria. [7]Department of Complex Systems, National Centre for Nuclear Research, Otwock-Świerk, Poland. [8]Institute of Applied Computer Science, Jagiellonian University, Kraków, Poland. [9]High Energy Physics Division, National Centre for Nuclear Research, Otwock-Świerk, Poland. ✉e-mail: eryk.czerwinski@uj.edu.pl

should respect the symmetries of its constituents, including discrete symmetries involving parity P, charge conjugation C and time reversal T invariance[4]. Fundamental QED respects P, C, T symmetries as well as the combinations CP and CPT. This paper presents the world's most precise test of T, P, and CP invariance in o-Ps decays. The measurement is realized by a method using the polarization of photons from o-Ps decays.

For a single electron or positron, C and CPT are seen to be working to 1 part in $10^{12}$ in their anomalous magnetic moments $a_e = (g - 2)/2$ (see refs. [5],[6]). The symmetry between electrons and positrons is also manifested in comparison of their masses $(m_{e^+} - m_{e^-})/m_{\mathrm{average}} < 8 \times 10^{-9}$ and electric charges $|q_{e^+} + q_{e^-}|/e < 4 \times 10^{-8}$[7]. CPT is a general property of relativistic quantum field theories beyond these charged leptons. A further recent test is the measurement of the antiproton-to-proton charge-mass ratio resulting in a 16-parts-per-trillion fractional precision in CPT invariance[8].

For CP, important information comes from electron electric dipole moment (eEDM). The tiny value $|d_e| < 4.1 \times 10^{-30}\,e\,\mathrm{cm}$[9] (see also refs. [10]–[12]) constrains the scale of any new CP violating interactions coupling to the electron. If such interactions couple with similar strength to Standard Model particles, then one finds constraints on the heavy particle masses similar to the constraints from the Large Hadron Collider at CERN. If, instead, the new interactions should involve ultralight particles, then one finds that their couplings to the electron should be less than about $\alpha_{\mathrm{eff}} \sim 5 \times 10^{-9}$ (see ref. [4]). Some new CP violation from beyond the Standard Model is needed to explain baryogenesis[13]—hence the interest in looking for such couplings. There are hints for possible CP violation in the neutrino sector though conservation is still allowed at the level of $1-2\,\sigma$[7],[14],[15].

Based on the EDM constraints, one expects CP to be working in Ps decays down to branching ratios at least about $10^{-9}$ (see ref. [4]). This has been explored in studies of CP-odd correlations[16], e.g., between final state photon momenta and the spin of the Ps. These experiments used ortho-positronium which decays into three photons with a lifetime in vacuum of 142 ns[17], and measured the correlation

$$O_1 = (\mathbf{S} \cdot \mathbf{k}_1)(\mathbf{S} \cdot (\mathbf{k}_1 \times \mathbf{k}_2)) \qquad (1)$$

with $\mathbf{S}$ the o-Ps spin vector and $\mathbf{k}_i$ the momenta of the emitted photons defined with magnitude $k_1 > k_2 > k_3$, and found the result $\langle O_1 \rangle = 0.0013 \pm 0.0022$ (see ref. [18])—consistent with zero at the level of $2 \times 10^{-3}$. Since Ps freely decays in vacuum to massless photons, it is not an eigenstate of T. This means that one can get CP, T and CPT violation mimicking final state interactions with magnitudes only detectable at the prevision level of about $10^{-9}$-$10^{-10}$ (see ref. [19]), beyond the scope of the present experiments.

In this paper, we develop a methodology made possible using the J-PET tomograph in Kraków using polarizations of the emitted photons, which are determined from Compton rescattering in the detector[20]. No magnetic field to control the o-Ps spin is needed in the experiment. The maximal cross-section of the Compton scattering is for the direction perpendicular to the electric field and polarization axis $\boldsymbol{\epsilon}$ of the incident photon[21],[22]. This leads to defining the polarization-related quantities

$$\boldsymbol{\epsilon}_i = \mathbf{k}_i \times \mathbf{k}'_i / |\mathbf{k}_i \times \mathbf{k}'_i|, \qquad (2)$$

where $\mathbf{k}_i$ and $\mathbf{k}'_i$ are the momenta of a photon from the positronium decay before and after Compton scattering in the detector, respectively[20]. These $\boldsymbol{\epsilon}_i$ vectors are most likely to be along the axis of the incident photon polarization vector and are even under P and T transformations.

One may then consider new correlations. Taking the polarization vector of one of photons $\boldsymbol{\epsilon}_i$ and momentum vector of another photon $\mathbf{k}_j$, we construct the momentum-polarization correlations[20]

$$O_2 = \boldsymbol{\epsilon}_i \cdot \mathbf{k}_j = \cos(\omega_{ij}) \qquad (3)$$

for all three independent combinations of these vectors, $(i, j) = (1, 2)$, $(1, 3)$, $(2, 3)$ with $\omega_{ij}$ being the angle between the $\boldsymbol{\epsilon}_i$ and $\mathbf{k}_j$ vectors. This correlation $O_2$ is odd under P, T and CP transformations. If the expectation value of $O_2$ does not vanish, then each of T, P, and CP symmetries would be violated in the o-Ps decay. Measurement of the correlation $O_2$ can be performed without an external magnetic field and without control of the o-Ps spin.

Here, we present an investigation of discrete symmetries in the o-Ps system based on the momenta and polarizations of the emitted photons Eq. (3) in o-Ps decays over the entire range of $\omega_{ij}$:

$$\langle O_2 \rangle = \langle \cos(\omega_{ij}) \rangle = \langle \boldsymbol{\epsilon}_i \cdot \mathbf{k}_j / k_j \rangle. \qquad (4)$$

We calculate $\langle O_2 \rangle$ from a distribution which is the sum of all independent combinations of $\omega_{ij}$. We find a value consistent with zero at 68% confidence level, as expected from the underlying QED with a threefold precision improvement over the previous measurements of the CP-odd correlation, Eq. (1), where the o-Ps spin was used to define the correlation. The bound state o-Ps decays obeys the CP symmetry of the underlying QED dynamics. Here one is probing the discrete symmetry properties of QED. Weak interaction effects are characterized by a factor $G_F m_e^2 \approx 10^{-11}$ with $G_F$ the Fermi constant, and would only be manifested with very much enhanced precision.

## Results
### Detector
The strategy we use here is to study the discrete symmetries associated with the operator correlation, Eq. (3), involving the momenta of photons from the o-Ps decay and the photon polarization-related vectors $\boldsymbol{\epsilon}_i$, which are measured using the Jagiellonian Positron Emission Tomograph (J-PET)[22-25]. The J-PET detector is based on plastic scintillators and is designed for total body scanning[26] in medicine[25,27,28] as well as biomedical studies[29,30] and fundamental physics research[4,20,22]. The J-PET detector is described in more detail in "Methods". For the measurement reported here, the positrons are emitted from a radioactive $^{22}$Na source placed at the center of the detector (Fig. 1a).

The source is coated with a porous polymer material to increase the probability of o-Ps creation[31] and inserted in the small vacuum chamber to decrease the background contribution from positron annihilation in the air. The annihilation photons from o-Ps → 3$\gamma$ are registered in three layers of scintillator strips forming a barrel shaped detector (Fig. 1a). Detection of annihilation photons in a given scintillator is based on registration of a light pulse at both ends of the scintillating strip. The light is collected by means of attached photomultipliers and the interaction, later on referred to as hit, is confirmed if signals at both ends of the strip are over a 30 mV threshold within a coincidence time of 6 ns. Figure 1b presents an example of a signal event of o-Ps → 3$\gamma$ annihilation for the CP symmetry test. The novelty of the reported measurement is in the determination of the polarization plane of annihilation photons and the experimental coverage of the whole angular range of the tested correlation. In addition, application of data acquisition system based on a fast, trigger-less, field-programmable gate array (FPGA)[32,33] and good timing properties of the plastic scintillators used in the experiment[34-36] (short light signals with 1 ns rising and 2 ns falling edges) and a high activity $\beta^+$ radioactive source allowed us to register the highest number of o-Ps → 3$\gamma$ annihilations for discrete symmetry studies so far recorded.

In addition, the achieved high data throughput make it possible to use high granularity of the active detector elements. As a consequence the angular resolution in the plane perpendicular to the detector axis is

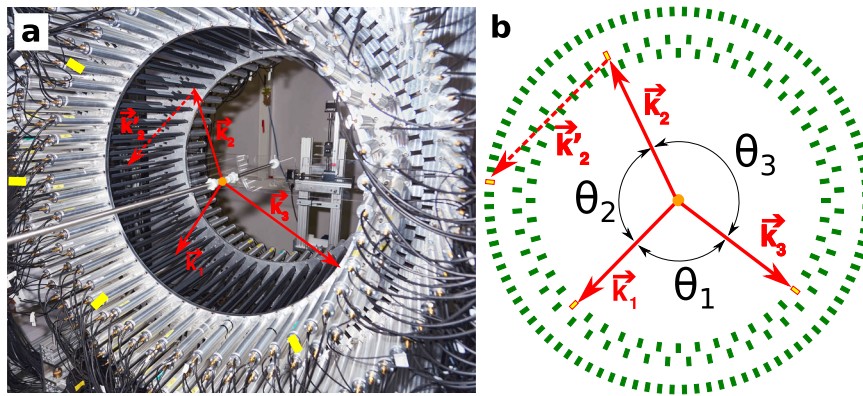

**Fig. 1 | The J-PET detection system.** The orange dot indicates the position of the sodium source. The superimposed solid red arrows indicate momenta of annihilation photons ($k_1 > k_2 > k_3$) originating from the decay of ortho-Positronium. The dashed red vector represents the momentum of the secondary scattered photon ($\mathbf{k'_2}$). Photomultipliers registering signals from these four photons are marked with yellow rectangles. **a** Photograph of the J-PET detector with the annihilation chamber installed at the center. Strips of plastic scintillator wrapped in black foil are mounted between two aluminum plates. Photomultipliers reading optical signals from these strips are inserted in aluminum tubes with mu-metal insert for optic and magnetic isolation. **b** Scheme of the J-PET detector where scintillators are drawn as green rectangles. For every selected event the directions of the momentum vectors for the three annihilation photons are reconstructed between the known position of radioactive source and the reconstructed hit point. Due to momentum conservation these three vectors are co-planar (annihilation plane). In the presented example the photon with medium energy ($\mathbf{k_2}$) interacts with the detector material and scatters as $\mathbf{k'_2}$ (forming the scattering plane). The angles between photon momenta are indicated such as $\theta_1 < \theta_2 < \theta_3$. Note that ordering of these angles is not directly related to the ordering of the momenta.

0.5°, which is important for the determination of the momentum direction.

## Signal and background

In order to construct the operator correlation defined in Eq. (3), three vectors of photon momenta are required: the momentum vectors of a Compton scattered photon before and after scattering, and an arbitrarily chosen one of two remaining photons from the o-Ps annihilation. However, for proper o-Ps → 3γ event identification and $k_1 > k_2 > k_3$ ordering, the momenta of all photons from o-Ps decay must be reconstructed. In this work we consider the expectation value of the distribution of the sum of three independent operators constructed with the aforementioned vectors. The momentum vector of a photon is reconstructed on the basis of its origin point and point of interaction with the scintillator. The origin point is common for photons emitted from the o-Ps decay and is equivalent to the source position. The point and time of interaction with the scintillator are calculated on the basis of the difference and sum of times, respectively, of registered signals at both ends of the scintillator strips[23]. The main experimental background to o-Ps → 3γ signal events (described in detail in "Methods") consists of (i) p-Ps → 2γ events with single scattering registered, (ii) events with multiple scattering of a single photon between active elements of detector, and (iii) cosmic rays.

## Analysis scheme

A signal event consists of four depositions of energy inside scintillating strips: three from o-Ps → 3γ and one from registered Compton scattering. Hits with energy deposition of at least 31 keV are registered by the data acquisition system, DAQ. Background interactions from p-Ps → 2γ and cosmic radiation with high energy depositions are suppressed by the requirement of time-over-threshold, TOT, less than 17 ns. Hits from o-Ps → 3γ decays (for each combination of three hits within an event) are identified based on the angular correlation between annihilation photons, comparison of their emission time and coplanarity of the momentum vectors of the annihilation photons. The energy of the annihilation photons is calculated from the angular dependence between all three photons from the o-Ps decay[37] and the momenta are ordered as $k_1 > k_2 > k_3$. Finally, the assignment of one of the remaining hits in the event to one of the photons originating from o-Ps → 3γ is based on the smallest value of the difference between calculated and measured time of flight of photon between $\mathbf{k_i}$ and $\mathbf{k'_i}$ interactions. The detailed description of the applied selection criteria is in "Methods". After the aforementioned selection", the final sample consists of $7.7 \times 10^5$ events. The angular correlation between momenta of annihilation photons for the final sample is presented in the Fig. 2a.

## Expectation value of the correlation $O_2$

The distribution of the reconstructed correlation defined in Eq. (3) is presented in the Fig. 2b. For the first time, the whole range of the CP asymmetry operator is measured. For the distribution of $O_2$ operator, the background expected on the grounds of performed Monte Carlo simulations is subtracted from the experimental distribution. The resulting distribution is corrected for the detector acceptance and analysis efficiency. The expectation value of the operator correlation $O_2$ is determined to be

$$\langle O_2 \rangle = 0.0005 \pm 0.0007_{\text{stat.}} \quad (5)$$

The systematic error contributions to this result are estimated from hit spatial, temporal and energy resolutions. The possible influence of cosmic rays is tested on the basis of a dedicated measurement without the positron source, but with an identical data processing scheme to that used for the $\langle O_2 \rangle$ determination. No significant systematic error from any contribution is found. The expectation value of the operator $O_2$ is consistet with zero within achieved accuracy, therefore no P, T, and CP asymmetry is observed.

## Discussion

Our methodology using the polarization of photons from positronium decays has allowed us to make the world's presently most accurate test of CP symmetry in o-Ps decays. The experiment uses the polarizations of photons emitted in the decay measured through the non-local correlation in Eq. (3), which is independent of the o-Ps spin. It involves the o-Ps decay and rescattering in the detector. Previous tests of CP-odd[18] and CPT-odd[38] decays of o-Ps were conducted by measuring angular correlations between momenta of the annihilation photons and the spin of the ortho-positronium only at specific fixed angles. Recently the J-PET group improved the test of CPT symmetry by measuring a momentum-spin correlation with full angular coverage[39]. The CP result reported here is also obtained using the full kinematic

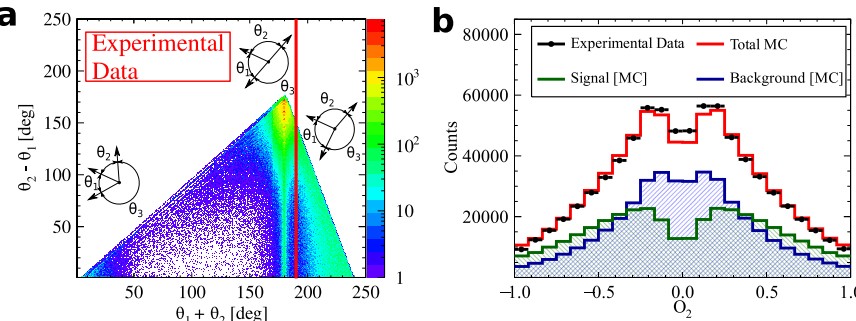

**Fig. 2 | Composition of the experimental data sample. a** Distribution of the sum and difference of the two smallest angles between photon momenta ($\theta_1 < \theta_2 < \theta_3$). The superimposed black pictographs present three different orientations of the momentum vectors for multiple scattered events (bottom left region), p-Ps → 2γ events with single scattering (vertical band around $\theta_1 + \theta_2 = 180°$) and o-Ps → 3γ signal events (bottom right region). The red vertical line corresponds to a $\theta_1 + \theta_2 \geq 190°$ cut applied for the signal selection. **b** Measured distribution of

asymmetry operator Eq. (4) for experimental data (black circles) and simulated histograms for signal (green), background (blue) and combined signal and background (red). The discrepancy between simulated distribution and data points for the two central bins may be explained by the rapid change of efficiency distribution in that region, but this effect is negligible comparing to the achieved accuracy of the final result.

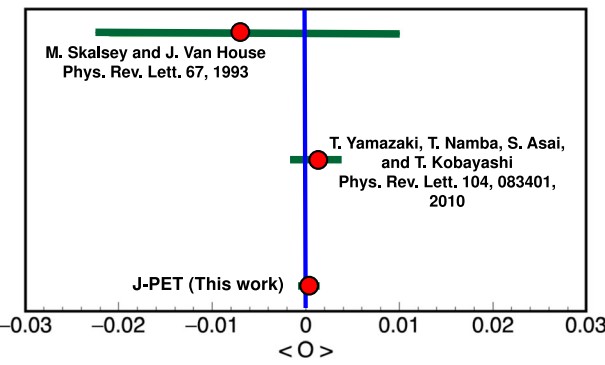

**Fig. 3 | Summary of searches for CP-odd ortho-Positronium decays.** The two upper results[18,53] are performed for the operator correlation $O_1$ defined in Eq. (1), whereas J-PET is using the new correlation $O_2$ constructed with the polarization vector in Eq. (3). The blue vertical line indicates no CP symmetry violation, while the green bars for each measurement correspond to the total uncertainty calculated as statistical and systematic uncertainties combined in quadrature.

range of photons appearing in the correlation $O_2$. With our method, the CP test is performed without the need to control the o-Ps spin using an external magnetic field. It is the first simultaneous test of P, T, and CP symmetries using the angular correlation between momentum of one of the annihilation photons and the polarization plane of another annihilation photon. The result reached the precision of $\mathcal{O}(10^{-4})$, which represents a threefold improvement in the search for CP-odd decays of o-Ps (Fig. 3). The use of polarization of photons for correlations like $\boldsymbol{\epsilon} \cdot \mathbf{k}$ or $\boldsymbol{\epsilon} \cdot \mathbf{S}$, where $\mathbf{S}$ is the spin of the positronium, opens a new class of discrete symmetry tests in positronium decays[20].

The new result might be further improved using the methods introduced here together with upgrades in the J-PET detector. These experiments will be conducted with a modular J-PET detector having about 20 times higher sensitivity for the registration of ortho-positronium. The modular version of the J-PET system[40] with increased acceptance is currently being used for a measurement of the P, T, CP, and CPT symmetries with a goal of reaching $10^{-5}$ accuracy.

## Methods
### Experimental setup
The J-PET shown in Fig. 1, is a multi-purpose, axially symmetric detector in the form of a barrel constructed with three layers of plastic scintillator strips[23,41,42]. Two inner layers consist of 48 strips each, placed at 425-mm and 467.5-mm radius, respectively, with the second

layer rotated by 3.75° with respect to the first one. The outer layer is composed of 96 strips at radius 575 mm. A single strip of J-PET is $500 \times 19 \times 7$ mm³ and made of fast-timing plastic scintillator[23,34,41,42] wrapped with two kinds of foils: external (for optical isolation) and an internal one to reflect the light from the scintillator. The position of a photon interaction along a scintillator strip is derived from the time difference of signals from two photomultipliers attached to a given strip, whereas the time of interaction is calculated from a sum of times of these signals. Each $19 \times 7$ mm² side is optically connected to the R9800 Hamamatsu photomultiplier[23,24]. Signals from 192 photomultipliers are probed in voltage domain at four different amplitude thresholds (Fig. 4). In total, up to eight measurements of time (leading and trailing edges) are performed allowing for precise signal start time derivation and time-over-threshold measurement equivalent to the photon deposited energy determination[32,43,44]. In the reported measurement, the amplitude thresholds are equal to 30, 80, 190, and 300 mV. All Time-to-Digital Converter (TDC) channels are distributed on the eight Trigger Readout Boards (TRBs) in the trigger-less manner[33]. Large amounts of data are registered due to the trigger-less data acquisition system DAQ[33], namely for the 1 MBq source there are $10^5$ hits per second collected, which translates into 28 MBps of recorded data. The Lempel–Ziv–Markov chain algorithm[45] is used to compress the data. The 122 days of data taking reported here resulted in 100 TB of archived data. For long-term storage, data were recorded on magnetic tapes in Linear Tape-Open version 7 (LTO-7). Data analysis of J-PET files was performed with a dedicated analysis framework[46,47]. In the reported measurement, four data campaigns were carried out: two with $^{22}$Na source of 5 MBq activity and two with activity of 1 MBq. The source was inserted between two 3-mm thick pads of XAD-4 porous polymer[48] and placed in the center of PA6 polyamide cylindrical chamber of inner diameter of 10 mm located on the axis of the J-PET detector (Fig. 1a). A vacuum system connected to the source holder ensured a pressure at a level of -1.5 × 10⁻⁴ Pa inside its volume. Taking into account the density of the chamber material (1.14 g/cm³), the thickness of the outer wall (1 mm), and mass attenuation coefficient[49], the attenuation of photons from o-Ps annihilation is estimated to 1%.

For Monte Carlo simulations, the geometry and material of the annihilation chamber and active detector elements (scintillator strips) are implemented in the GEANT4 toolkit[50]. The experimental resolution of the whole experimental setup is introduced as Gaussian smearing with standard deviation σ. For the deposited energy $\sigma_E$=14 keV, for time of the hit $\sigma_T$ = 225 ps and for Z-position of the photon interaction $\sigma_Z$ = 2.4 cm. The values of the above-mentioned smearing parameters are obtained from the fit of Monte Carlo distributions to data points shown in Fig. 5.

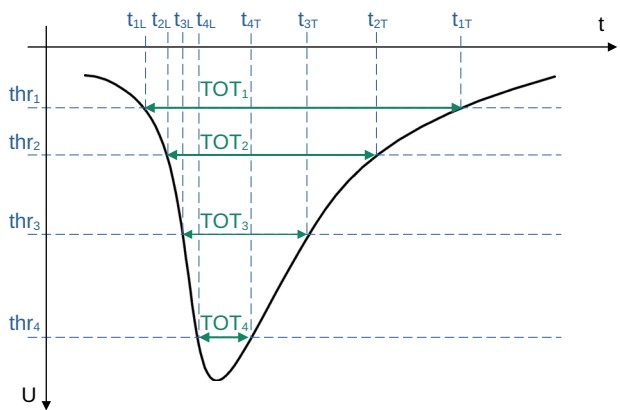

**Fig. 4 | The idea of a TOT measurement for constant thresholds *thr$_i$*, where *i* =1, 2, 3, 4.** A negative electric signal (black line) is probed in the voltage domain *U* at four voltage thresholds allowing for determination of crossing time *t*, with leading $t_{iL}$ and trailing $t_{iT}$ edge of the signal. The energy carried by the signal is therefore proportional to the area under the signal, which is estimated as a sum of areas of rectangles limited by neighboring thresholds and registered times. The energy deposited by a photon is proportional to the light collected by photomultipliers at both ends of the scintillator strip. The TOT for a given deposition (hit)[32,43,44] is calculated as the normalized sum of products of a difference of consecutive thresholds with respect to the baseline thr$_0$ = 0 mV and TOT measurements at both ends of the scintillator strip for each threshold. The normalization factor is the difference between two highest thresholds, namely $\text{TOT} = \sum_{j=1}^{2} \sum_{i=1}^{4} \text{TOT}_i^j \cdot (\text{thr}_i - \text{thr}_{i-1})/(\text{thr}_4 - \text{thr}_3)$, where *j* counts TOT measurements at both ends of the scintillator strip.

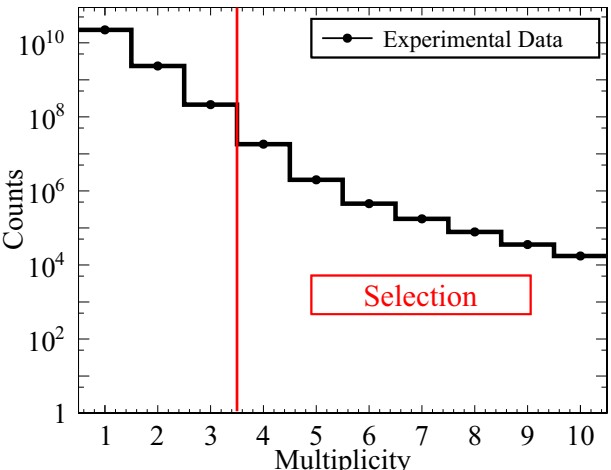

**Fig. 6 |** Exemplary distribution of hit multiplicity within events.

## Signal candidates selection

A signal event is an o-Ps → 3$\gamma$ decay with one of the annihilation photons undergoing Compton scattering. Therefore a signal candidate consists of four registered hits of photons in scintillator strips: three coming directly from annihilation of o-Ps and one as a secondary scattered photon.

The selection of signal candidates is a three-step process:

1. Selection of at least four candidate hits within one event (Fig. 6), where each hit fulfills the following conditions:

   the energy deposited in the scintillator must not be smaller than 31 keV to reject multiple scattered hits (this value corresponds in fact to the lowest threshold set at DAQ);

   the position of the hit at the scintillator strip must be within ± 23 cm window around the center point to suppress hits at the ends of scintillators due to scatterings from aluminum plates holding scintillator strips (Fig. 1a);

   the registered TOT value must be ≤17 ns to reject hits originating from cosmic radiation (tested with separate data-taking campaign without the radioactive source) and to reduce the p-Ps → 2$\gamma$ background component (Fig. 7a), as well as the deexitation photon from $^{22}\text{Ne}^*$;

2. The identification of hits from o-Ps → 3$\gamma$ decay was performed as follows:

   the emission time was calculated for each hit as a difference between the registered time (hit time) and a travel time (ratio of the distance between source and hit position and speed of light);

   the emission time spread (ETS) was calculated as a difference between last and first emission time of three candidates; this ETS must be less than or equal to 1.4 ns to ensure that hits originate from the same o-Ps decay (Fig. 7b);

   for a source position of $(s_x, s_y, s_z)$ a distance between annihilation plane (spanned by the annihilation photons' momenta and defined as $Ax + By + Cz + D = 0$) and the source was calculated as $\text{DOP} = |A \cdot s_x + B \cdot s_y + C \cdot s_z + D| \cdot (A^2 + B^2 + C^2)^{-\frac{1}{2}}$; the DOP constructed with three candidate hits must be less than or equal to 4 cm to reject hits from multiple scatterings (Fig. 7c);

   at the decay plane the sum of the two smallest angles between photon momentum vectors from o-Ps → 3$\gamma$ decays must be greater than or equal to 190° (Figs. 2, 5, and 8) to reject main contribution from p-Ps → 2$\gamma$ events with multiple scattered photons;

   for events with more than three hits a combination with the smallest (ETS)$^2$ + (DOP)$^2$ value was selected;

3. After the above-mentioned selection of three photons from o-Ps → 3$\gamma$ decay, the assignment of one of the remaining hits in the event as the interaction of a scattered photon from the o-Ps → 3$\gamma$

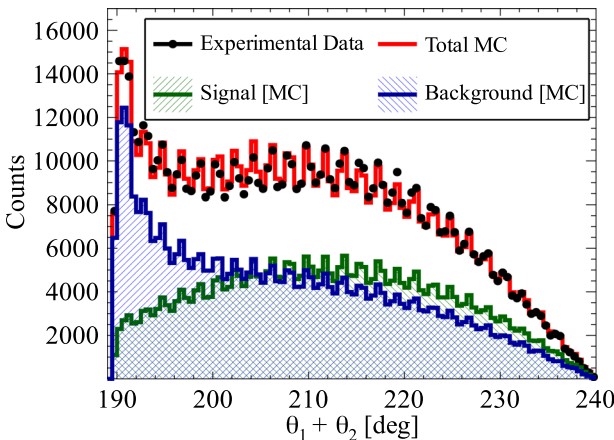

**Fig. 5 | Distribution of the sum of the two smallest relative azimuthal angles ($\theta_i$ and $\theta_j$) between the registered annihilation photons (projection on the horizontal axis of experimental data from Fig. 2a and simulation of background and signal events (Fig. 8b, c, respectively).** Experimental data points are marked with black circles, while histograms represent the results of reconstructed Monte Carlo simulations for signal (green), background (blue), and combined signal and background simulated distributions (red). The experimental histogram contains all the events after the analysis. Visible multiple maxima and minima are due to distances between scintillating strips (Fig. 1b). The main maximum at 190° is in fact a remaining tail of background component of the p-Ps → 2$\gamma$ process where $\theta_i + \theta_j = 180°$ and one of the photons undergoes a single scattering.

The *X* and *Y* coordinates of photon interactions are generalized to the center of a given scintillator strip. In order to reduce the statistical fluctuation of simulated samples of events the generated signal and background events are 3.5 and 2.4 times bigger than the contributions found in the experimental data, respectively.

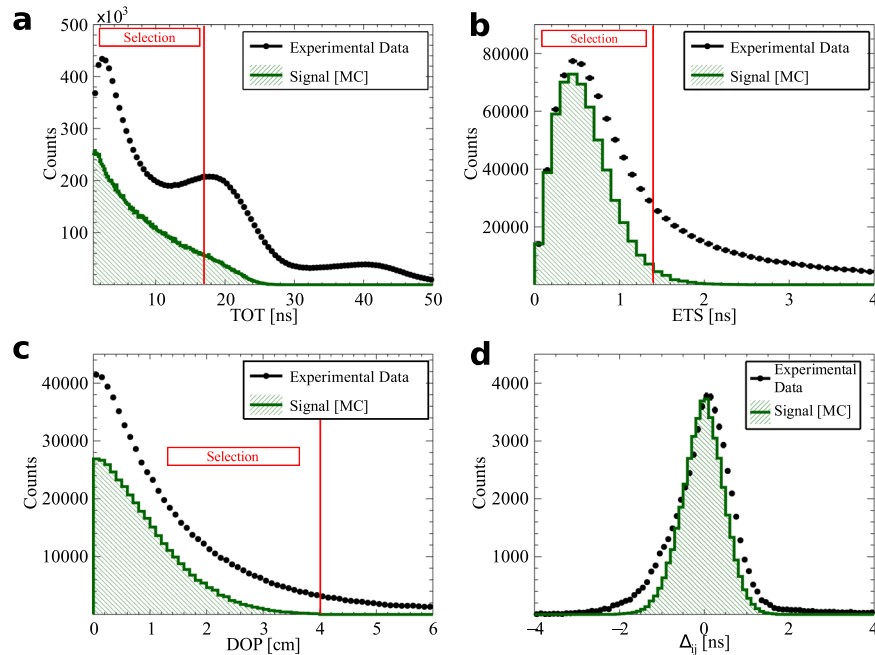

**Fig. 7 | Exemplary spectra for the selection criteria with the superimposed red line of the applied cut value.** The definition of each variable is given in the text. **a** Time-over-threshold TOT for each hit for events with multiplicity greater than 4. The o-Ps → 3γ candidate is constructed out of 3 hits with the smallest value of

$(ETS)^2 + (DOP)^2$. The final candidates are selected after surviving cuts on ETS (**b**) and DOP (**c**). **d** The assignment of the scattered photon to one of o-Ps → 3γ candidates is based on the smallest scatter test value.

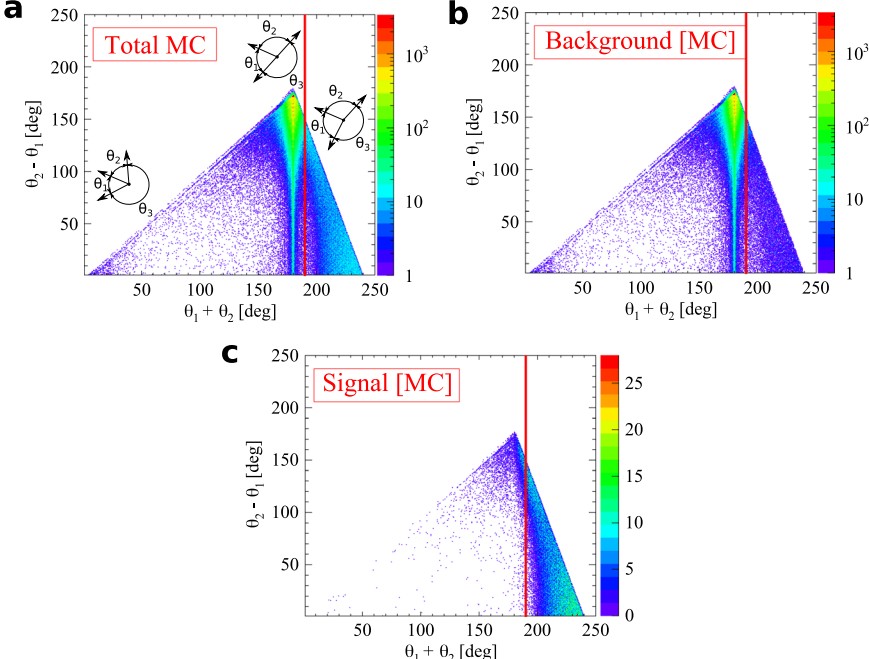

**Fig. 8 | Identification of background to o-Ps → 3γ signal events.** $\theta_1$ and $\theta_2$ indicate two smallest relative angles between momentum vectors of photons. The super-imposed black pictographs at the first plot present three different orientation of momentum vectors for events with multiple scatterings (bottom left region of each plot), p-Ps → 2γ events with a single scattering (vertical band around $\theta_1 + \theta_2 = 180°$)

and o-Ps → 3γ signal events (bottom right region). The red vertical line corresponds to a $\theta_1 + \theta_2 \geq 190°$ cut applied for signal selection. **a** Full sample of MC simulated events. **b** Background events within the Monte Carlo sample. **c** Simulated signal events.

decay was based on the smallest time difference $\Delta_{ij} = (t_j - t_i) - |\mathbf{r}_j - \mathbf{r}_i|/c$ between the reconstructed and expected time of flight of the scattered photon, where the measured time and position of interactions are $t_i, \mathbf{r}_i$ for the $i$th selected

annihilation photon and $t_j, \mathbf{r}_j$ for $j$th candidate for scattering of the $i$th photon, respectively ($i = 1, 2, 3$ and $j = 4, \ldots$, multiplicity), where multiplicity is the number of registered hits per event, see Fig. 7d.

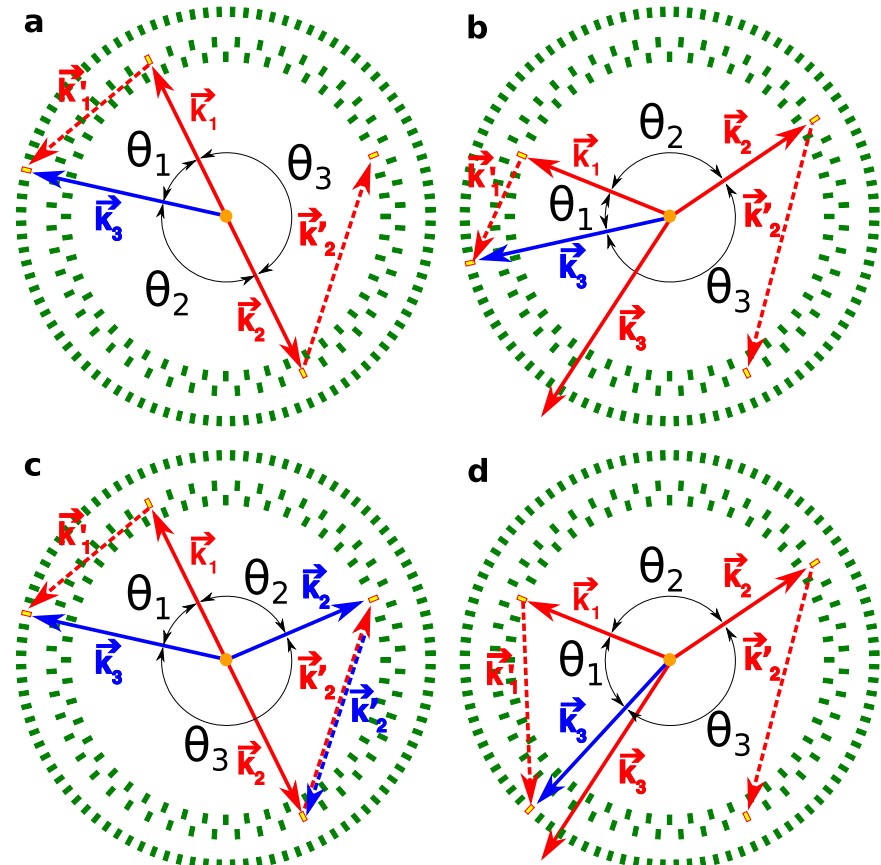

**Fig. 9 | Topology of background events.** Scintillators of J-PET are schematically presented as green rectangles. Scintillators registering the photons in presented events are indicated as yellow rectangles. Solid lines denote photons originating from Ps annihilation, while dashed lines represent scattered photons. Momenta of signal photons are drawn with red color, while incorrectly reconstructed ones - with blue. The following convention is used: $\theta_1 < \theta_2 < \theta_3$. **a** An exemplary event of p-Ps → 2γ decay with the registration of both annihilation photons and two scatterings. The one with direction of $\mathbf{k}'_2$ is correctly recognized during analysis as a scattered hit, while the scattered $\mathbf{k}'_1$ is wrongly assigned as $\mathbf{k}_3$. **b** A misreconstructed o-Ps → 3γ decay due to wrong assignment of the scattered hit. The photon from o-Ps

annihilation marked as $\mathbf{k}_3$ (red) is not detected, while $\mathbf{k}_2$ scatters as $\mathbf{k}'_2$ and is properly reconstructed. The annihilation photon with momentum direction marked as $\mathbf{k}_1$ also scatters. It is not reconstructed as $\mathbf{k}'_1$, but incorrectly reconstructed as annihilation photon $\mathbf{k}_3$ (blue). Both events presented in the top row would be rejected by the $\theta_1 + \theta_2 \geq 190°$ criterion, while events from the bottom row would be incorrectly accepted. **c** p-Ps → 2γ decay with registration of both annihilation photons and two scatterings, but only one of the hits is correctly assigned ($\mathbf{k}_1$). $\mathbf{k}'_1$ is misidentified as $\mathbf{k}_3$ (blue), while $\mathbf{k}'_2$ (red) is misidentified as $\mathbf{k}_2$ (blue) and $\mathbf{k}_2$ (red) as $\mathbf{k}'_2$ (blue). **d** An event similar to the one presented in the top right panel, but with a topology immune to the $\theta_1 + \theta_2 \geq 190°$ cut.

The main background contributions to o-Ps → 3γ events are p-Ps → 2γ events with the registration of additional scatterings, partially reconstructed o-Ps → 3γ decays mixed with different hits, and o-Ps → 3γ decays with wrong assignment of hits to photons from annihilation and scattering. As an example, a composition of different background events is presented in Fig. 9. Differences between signal and background events were identified in the two-dimensional distribution of difference and the sum of relative angles between momentum vectors of photons (Fig. 8a for the full Monte Carlo sample, Fig. 8b for simulated background events, and for simulated signal events in Fig. 8c)[20].

All the values of the applied cuts were optimized for the best Monte Carlo to data agreement of the distribution of the sum of two smallest relative azimuthal angles between the annihilation photons (Fig. 5).

The number of generated Monte Carlo events exceeds the number of experimental events. Therefore, the normalization of Monte Carlo contributions was performed with two independent parameters: one for o-Ps → 3γ signal events and a second scaling parameter for remaining background events. The histograms in Fig. 5 are shown after the normalization procedure.

The geometrical acceptance of the J-PET detector is determined using Monte Carlo simulations. It is estimated as a ratio of the number

of simulated signal events to the number of generated events with the o-Ps decay into three photons and one scattered photon. The signal events are those in which three photons from the o-Ps to 3γ decay interacted in the detector, and at least one of them scattered a second time. The selected Monte Carlo signal events after the entire analysis chain are used for the determination of analysis efficiency as a ratio of the number of signal events surviving selection cuts over a number of registered signal events. The combined distribution of acceptance and analysis efficiency is presented in the Fig. 10a.

**Determination of expectation value**

Having three hits assigned to an o-Ps → 3γ decay and a fourth hit correlated as a scattering to one of the annihilation photons, $\cos(\omega_{ij})$ is determined for each event after the analysis selection chain. The energy of annihilation photons from o-Ps decay is calculated in the basis of their relative angles[37], and the momenta are ordered accordingly $k_1 > k_2 > k_3$. Then $\cos(\omega_{ij})$ is calculated according to Eq. (4) while $\epsilon$ is derived from Eq. (2). From the experimental distribution of $\cos(\omega_{ij})$ (Fig. 2b) the normalized background spectrum is subtracted. The obtained distribution is finally divided by Monte Carlo-derived distributions of the total efficiency (Fig. 10a) and a mean value of the distribution is calculated as the expectation value of the

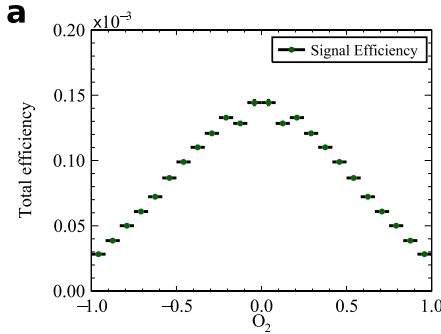
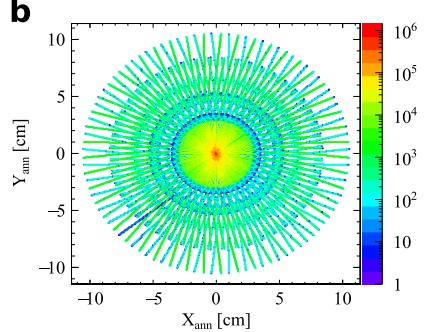

**Fig. 10 | The signal efficiency as a function of $O_2$ and the image of p-Ps → 2γ annihilation points. a** Monte Carlo simulation derived distribution of efficiency including all selection criteria applied for the described analysis and geometrical acceptance of the detector for signal events. **b** Annihilation points in the XY plane reconstructed with 2γ events from 19 days of measurement. The visible rosette pattern is due to the geometrical acceptance of the detector (placement of scintillators strips presented at Fig. 1).

correlation $O_2$, Eq. (3), along with the statistical error of the expectation value.

### Estimation of systematic uncertainties

Contributions from all selection criteria to the systematic uncertainty were calculated by changing the given cut value by its resolution and performing the whole analysis chain again. Following the approach proposed by Barlow[51,52] the statistical significance of the systematical contribution from each cut was calculated as the difference between the expectation value of operator $O_2$ obtained this way and the final result normalized to the uncertainty. The resolution of the distance to the annihilation plane (DOP) is estimated to be 1.1 cm. The position of the energy deposition by a photon along the scintillator strip $Z_{hit}$ is known to the accuracy of 2.4 cm. The angular resolution for $\theta_1 + \theta_2$ determination (Fig. 5) is 1.5°. The emission time spread ETS of photons originating from the o-Ps decay is about 0.5 ns. The TOT is measured with 1.2 ns resolution, while the DAQ registration threshold is known up to 14 keV. The position of the annihilation measured from p-Ps → 2γ decays is known to 0.5 mm accuracy in the $X$–$Y$ plane and 0.4 mm resolution along the Z axis. It is worth to mention that the annihilation place (source position) is continuously monitored with p-Ps → 2γ events, as an intersection of lines formed with two monoenergetic, back-to-back annihilation photons. Two-dimensional distribution of reconstructed annihilation points in the $XY$ plane is presented in the Fig. 10b. Possible influence of bin width at $\cos(\omega_{ij})$ spectrum (Fig. 2b) to the final result is tested with double and twice reduced width of the bin. Finally the contribution of pure cosmic rays is estimated with a separate measurement without positronium source for which registered data is analyzed the same way as in case of o-Ps → 3γ decays. The resulting distribution of $\cos(\omega_{ij})$ is subtracted from the experimental spectrum. In addition, for conservative consideration the cosmic rays distribution is added to the experimental spectrum. The result shows no statistically significant contribution from any of the aforementioned parameters.

### Data availability

The datasets collected in the experiment and analyzed during the current study are available under restricted access due to the large data volume. Direct access to the data can be arranged on request by contacting the corresponding author.

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

## Acknowledgements

The authors acknowledge the technical and administrative support of A. Heczko, M. Kajetanowicz and W. Migdał. This work was supported by the Foundation for Polish Science through the TEAM POIR.04.04.00-00-4204/17 program (P.M.), the National Science Centre of Poland through grants MAESTRO no. 2021/42/A/ST2/00423 (P.M.), OPUS no. 2019/35/B/ST2/03562 (P.M.) and SONATA BIS no. 2020/38/E/ST2/00112 (E. P.d.R.), the Ministry of Education and Science through grant no. SPUB/SP/490528/2021 (P.M.), the EU Horizon 2020 research and innovation pro-gramme, STRONG-2020 project, under grant agreement No 824093 (P.M.), and the SciMat and qLife Priority Research Areas budget under the program *Excellence Initiative - Research University* at the Jagiellonian University (P.M.), and Jagiellonian University project no. CRP/0641.221.2020 (P.M.).

## Author contributions

The experiment was conducted using the J-PET apparatus. The J-PET detector, the techniques of the experiment, and the sym-metry test involving polarization were conceived by P.M. The data analysis was conducted by J.R. Signal selection criteria were developed by P. M. and E. C., applied by J. R., and verified by E.C. Authors: P.M., E.C., J.R., E.Y.B., N.C., A.C., C.C., M. Dadgar, M. Das, K.D., A.G., B.C.H., K. Kacprzak, T. Kaplanoglu, Ł.K., K. Klimaszewski, P.K., G.K., T. Kozik, W.K., D.K., S.M., W.M., S.N., S.P., E.P.d.R., L.R., S.S., S.C., R.Y.S., M. Silarski, M. Skurzok, E.Ł.S., P.T., F.T.A., K.T.A., K.V.E., and W.W. participated in the construction, commissioning, and operation of the experimental setup, as well as in the data-taking campaign and data interpretation. M.G. and B.J. designed and constructed the positronium production chamber. S.N. and G.K. optimized the working parameters of the detector. K.D., A.G., K. Kacprzak, and W.K. developed the J-PET analysis and simulation framework. G.K. developed and operated the DAQ system. K.D., M. Silarski and M. Skurzok performed timing calibration of the detector. K. Klimaszewski, P.K., W.W., L.R., and R.Y.S. managed the computing resources for high-level analysis and simulations. E.C. developed and operated short- and long-term data archiving sys-tems and the computer center of J-PET. S.S. established relation between energy loss and TOT. S.D.B. provided advice for the the-ory. P.M. managed the whole project and secured the main finan-cing. The manuscript was prepared by P.M., E.C., S.D.B., and J.R. and was then edited and approved by all authors.

## Competing interests

The authors declare no competing interests.

## Additional information

Eryk Czerwiński.

