## [Peer Review File · Nature Communications]

Discrete symmetries tested at 10^{-4} precision using linear polarization of photons from positronium annihilationsREVIEWER COMMENTS

Reviewer #1 (Remarks to the Author):

Authors search for the CP violation in QED with the JPET detector which has a novelty and it is unique since the authors develop a new method using polarization of the emitted photons which are determined from Compton scattering in the detector which is different from a previous experiment which used the magnetic field for o-Ps spin polarization.

Authors set three times better limits than previous results if we just take an account statistical fluctuation only. However, I have a deep concern about the systematic uncertainty of MC since authors heavily rely on MC for the CP asymmetry calculation including signal, background, and efficiency estimation.

Even though results are important for the community, there are several major concerns that need to be resolved before more detailed comments.

1. Explain why author's novel method is better than the magnetic field method, are the current method could reduce systematic uncertainty compared with the previous method?
2. Fig. 2 is the most important plot for the CP asymmetry measurement. However, data and MC does not agree well. Make a plot with $\text{abs}(\text{data}-\text{MC})/\text{data}$ with % residual with respect to O2 variable to see how much % the difference between MC and data in each bin. Keep in mind that authors measure the CP asymmetry in 10^{-4} level, then data and MC need to agree at an excellent level which might be a serious issue.
3. To prove both signal MC and background MC agree well with data, authors need to show all CP asymmetry-related variable plots to prove that MC and data agree well. For example, angular, and vertex distribution need to be shown with data and MC overlapping like Extended Data Fig. 2. Extended Data Fig. 3 and Fig. 4 data need to be overlapped with signal+BG MC. Also, authors should show a plot of e_i distribution between k_2 and k_2' data overlap with signal+BG MC.
4. Why there are so much up and down fluctuation in histogram $\theta_1+\theta_2$ histogram? If it is caused by the histogram of the binning effect, the authors need to correct it. $\theta_1-\theta_2$ plot needs to be added too.

5. To see the sensitivity of CP in the JPET detector, authors need to perform toy MC (which is popular in search experiments in HEP) to make sure make a 0.001 to 0.0001 level of CP violation and check whether or not authors achieve the same violation as put it in with current statistics of data.

6. How you can verify acceptance as shown in Extend Data Fig. 7 of MC in 10^{-4} level

7. Explain detail how authors got $\langle O_2 \rangle = -0.0005 \pm 0.0007$ from Fig 2 Right plot. The authors obtained 0.0007 level of statistical uncertainty with 7.7×10^5 events and more than half of background events need to be subtracted.

8. Also authors need to consider MC statistics based systematic uncertainty science authors used are negligible but authors only used 3.5 and 2.4 times MC which will contribute $\sim 60\%$ of current statistical uncertainty. Thus systematic uncertainty by MC statistics is not negligible.

9. Title "Matter-antimatter symmetry tested at 10^{-4} precision" is misleading. First, the authors tested CP violation in QED sector not matter-antimatter symmetry directly. We never call Matter-antimatter symmetry violation in the weak sector even if a large CP violation in weak sector has been observed. If authors want to put the above title, authors need to prove less than 0.001 CP violation will lead to matter-antimatter asymmetry and how much asymmetry can be predicted. Authors need to write 0.0007 (1 sigma) with only statistical uncertainty included instead of the 10^{-4} level in the abstract. Also, the authors need to make it clear that authors only test CP violations in the QED sector which is predicted to be very small, not like CP violations in the weak sector.

10. In the abstract, the authors wrote "Positronium, the simplest bound state of an electron and positron, is of recent interest with discrepancies reported between measured hyperfine energy structure and theory at the level of 10^{-4} and up to 4.5 standard deviations." It is not much related to CP violation in QED that authors need to remove the above sentence in the abstract.

11. Authors need to write a 90% confidence level limit on CP violation in the QED sector.

Reviewer #2 (Remarks to the Author):

The paper tests P, T and CP, and thus CPT, symmetries between matter and antimatter using decays of positronium in a way independent of the measurement of the spin of the positronium, using the J-PET tomograph. They confirm preservation of CPT symmetry between matter-antimatter at the level of 10^{-4} , consistent with Quantum Electrodynamics (QED) expectations. This level of accuracy is claimed by the authors to be important, given recent anomalies observed in hyperfine structure spectroscopy measurements of positronium, leading to a 4.5 sigma discrepancies between experiment and theory (the latter being mostly non relativistic QED bound state theory).

In view of such anomalies, testing CPT symmetry independently at this level in the positronium system acquires an important meaning, and this experiment, together with the innovative approach of using the J-PET tomograph, constitutes an important platform for excluding the possibility that the aforementioned anomalies are due to a fundamental breakdown of CPT symmetry between matter and antimatter in this system.

Being a theorist, I do not have the expertise to judge the experimental details, however I can judge the importance of the motivation for this work, and, in view of the above comments, I believe the article meets the stringent criteria for being published in Nature communications.

The paper discusses in my opinion in a clear way background effects that could affect the conclusions. However, one aspect which I could not see it discussed, are the prospects for increasing the accuracy of such CPT tests beyond the 10^{-4} level. In my opinion some speculations in this direction would make the paper more complete, especially because of the importance of the subject.

In general, I consider the paper important to be published in Nature communications, provided the authors take into account my suggestion on remarking on the prospects for improved sensitivity, so as to test CPT symmetry in positronium, or other similar systems.

Reviewer #3 (Remarks to the Author):

This paper reports on a much improved test of CP in $Ps \rightarrow 3\gamma$ decays, where the polarization of one of the gammas is measured and combined with the momentum vectors of the gammas to form a CP-sensitive term that is symmetric in the case CP is conserved. Polarization of a gamma is measured by detecting a Compton scattered photon in the JPET device.

The analysis is generally sound and straightforward to follow and the result represents a major advance in sensitivity; thus the paper merits publication in principle. However a small number of questions are not clear in this reviewer's mind, and would deserve explaining in more detail:

- event selection: hit multiplicities >3 appear to be used; with larger hit numbers, combinatorics and worsened resolutions can be expected; has the analysis also been done for $n_{\text{hits}}=4$ events only? as a function of n_{hits} ? has an optimum n_{hits} been searched for?

- the source is not point-like, but rather somewhat extended (5mm radius, several mm in length, 1-3 mm thickness); while I can't think of any asymmetry that could result from this extensive volume (in which gammas can scatter), I wonder how this rather large material budget can affect the resulting distributions. One concern might be that 2γ decays (accompanied by background hits) could more easily enter the 4 hit candidate sample as scattering would reduce the 180 degree opening angle.

- Fig. 2 left shows the experimental data, while the MC ingredients are shown in the supplementary material; Fig. 2 right shows the O_2 variable for the same data sets. How is the normalization (pg. 16) carried out? I am concerned that the simulation/fit of signal and backgrounds very poorly reproduces the experimental distribution in the small O_2 region, and even more so that the experimental distribution appears to have an asymmetry with respect to the (symmetric) MC in the second (and to a much smaller extent, the third) bin (0.1-0.2, resp. 0.2-0.3). Given the invisible error bars on the experimental points (presumably lying within the circles), the discrepancy is highly significant...

- a related figure regards Fig. 2 left: what does the residual 2-d distribution look like of the MC are scaled according to Fig. 2 right and subtracted from the experimental data of Fig 2 left?

- is the O₂ distribution in Fig. 2 right corrected for detection efficiency (suppl. fig 7)? What causes the enhanced/suppressed structures (1st, 2nd, 3rd bins) in both of these distributions?

Additional minor questions are:

- pg. 9 coplanarity of photons is an important selection variable, but the experimental distribution (and the backgrounds) is not provided...perhaps something to add to the supplementary material

- page 9, bottom: Do you mean that the Monte Carlo background is subtracted from the experimental distribution? The sentence reads ambiguously and would better be inverted.

- pg. 10: what do you mean by "complex" in "Positronium is the simplest complex bound state"? Isn't it the simplest bound state, together with hydrogen?

- pg. 14, second bullet: what is the energy of the ²²Ne deexcitation photon?

- the concept of ETS (and that of DOP) is not explained clearly (pg. 14 and extended data fig. 4) - presumably, the time of flight is calculated from the position of the detected hit for each photon, and the time of emission of the three photons reconstructed for a source assumed to lie at (0,0,0)? An equation or a sketch might help.

- have you developed a better proxy for photon energy via the use of multiple threshold TOT_s (suppl. fig. 1)? How does the calculation of the photon energies from the overall geometry of their emission directions and the assumption of $\text{Sum}_E = 2m_e$ compare with such an optimized proxy? What is the resolution of E_{gamma} when the approach of pg. 9 is used? Does the resolution of 2-gamma event TOT's match that of the background sample in the 4-gamma distribution of suppl. data fig. 4 top left?

In addition to the above, on a large number of occasions, English awkwardness (missing particle, formulation) is apparent in the text; given their number, listing all of them would be excessive, but in many cases "the" or "a" is missing, or sentences would benefit from being rewritten by a native speaker.

Below we answer point by point to the Reviewers' remarks. The modifications of the manuscript itself are marked with red. The comments of the Reviewers are quoted in italics for convenience.

Answers to Reviewer #1

We appreciate the time and effort you dedicated to review our manuscript. Thank you for your careful reading and prompt review. We are honored that you consider our measurement to be unique, and we greatly appreciate the valuable feedback you have provided. We believe that answers we provided support our result and the manuscript would be accepted for publication.

Authors search for the CP violation in QED with the JPET detector which has a novelty and it is unique since the authors develop a new method using polarization of the emitted photons which are determined from Compton scattering in the detector which is different from a previous experiment which used the magnetic field for o-Ps spin polarization. Authors set three times better limits than previous results if we just take an account statistical fluctuation only. However, I have a deep concern about the systematic uncertainty of MC since authors heavily rely on MC for the CP asymmetry calculation including signal, background, and efficiency estimation.

We are happy that the novelty of the performed measurement with three times better limits than previous results is recognized and the value of the method using polarization of the emitted photons is noticed. As described in our manuscript the systematical contribution was estimated following the approach proposed by Barlow (Ref. [52] and [53] in the revised version) by comparing the final value of the operator O_2 with the results obtained by changing the parameters of the measurement by their errors and the result shows no statistically significant contribution from any of the considered effects. We find the method described by Barlow (Ref. [52] and [53] in the revised version) as appropriate for the systematic error estimation. This method is also used by the other research groups as e.g.

- Lees, J.P. et al. Search for $B^+ \rightarrow K^+\tau^+\tau^-$ at the BaBar Experiment, Phys. Rev. Lett. 118, 031802 (2017).
- Achasov, M.N. et al. Search for the Process $e^+e^- \rightarrow \eta'\gamma$ with the SND Detector, Phys. Atom. Nuclei 83, 714–719 (2020).
- Aubert, B. et al. Search for Lepton Flavor Violating Decays $\tau^\pm \rightarrow l^\pm\omega$, Phys. Rev. Lett. 100, 071802 (2008).

Indeed, we use the Monte Carlo simulations for understanding the detector performance and for the efficiency corrections. However, it is important to stress that the raw, not corrected result is also showing no CP violation and is consistent with the final corrected result. Therefore, the final result does not depend on the corrections. The detector system

was especially designed to be symmetric such that in principle it does not introduces artificial asymmetries. This is because each out of 192 scintillator strips contributes to the registration of all O_2 values ("configurations"). Moreover, all configurations of the o-Ps $\rightarrow 3\gamma + \gamma_{scatter}$ are measured simultaneously without a need to change detector configurations. This is discuss more detailed below, in the answer to comments no. 2. and 6.

Even though results are important for the community, there are several major concerns that need to be resolved before more detailed comments.

1. *Explain why author's novel method is better than the magnetic field method, are the current method could reduce systematic uncertainty compared with the previous method?*

The method which we introduced in this article is different than the methods used so far allowing to study the CP symmetry by different class of operators constructed from momentum and polarization of photons. Thus this method enables the research of CP using new class of operators. In this sense it is not better but it is opening new possibilities for the CP symmetry tests. However, it is better than previously used methods as regards the control of experimental conditions. In the previous measurements the external magnetic field was used to control the spin orientation of the o-Ps. One of the improvement by not using the magnetic field is a lack of the magnetic field itself. In this case there is one less parameter to be controlled and included in the final uncertainty, eg. see the previously best result of Yamazaki et al. (Ref. [18]) where the uniformity of the magnetic field was estimated to about 10% over the volume of positronium production. Second reason is an absence of the magnets in the experimental setup, meaning no empty holes in the geometrical acceptance. The result of Yamazaki et al. (Ref. [18]) was achieved with fixed relative position of the detectors used for registration of annihilation γ . As a consequence only a specific configuration of the studied operator O_1 was measured. In case of the experiment performed with J-PET detector the full phase space of the O_2 operator is available (right plot of Fig. 2 and left plot of Fig. 10).

2. *Fig. 2 is the most important plot for the CP asymmetry measurement. However, data and MC does not agree well. Make a plot with $abs(data-MC)/data$ with % residual with respect to O_2 variable to see how much % the difference between MC and data in each bin. Keep in mind that authors measure the CP asymmetry in 10^{-4} level, then data and MC need to agree at an excellent level which might be a serious issue.*

As mentioned in the previous paragraph one of the novelties of our measurement is an access to the full phase space of the operator O_2 . Therefore the performed test of the CP symmetry is based on the full spectrum of the operator values and, as such, the expectation value is equivalent to the mean value of distribution of O_2 . The achieved sensitivity is due to acquired statistics as well as the total range of the operator values. The plot requested by the Reviewer #1 is presented on

the left plot of Figure A. There are two crucial points for the remark given by the

Figure A: **Left:** Plot of percentage residual for O_2 operator defined as $\text{abs}(\text{data-MC})/\text{data}$. **Right:** Difference between bins of O_2 operator (from the left plot) of negative and positive values, respectively. The black line is to guide the eye only.

Reviewer #1: estimation of the distribution of acquired signal events (by subtraction of MC background events from the data distribution) in the function of O_2 and the overall agreement between MC and data distribution in the function of O_2 . The key factor here is that in order to avoid vicious circle the Monte Carlo parameters, like background and signal normalization factors, were estimated at independent distribution (Fig. 5). Therefore, this distribution in Fig. 5 presents better MC to data agreement with respect to Fig. 2. These normalization factors were later on used for MC distributions of Fig. 2. The biggest discrepancies between MC and data (at the plot requested by the Reviewer #1) are for the side and central regions of O_2 values. It is important to stress here, that due to the nature of the O_2 operator itself the number of events tends to zero for $O_2 = 0$. Following Eqs. 2-4 from the manuscript, $O_2 = 0$ when the scattering occurred under 0° angle (no scattering) and therefore the determined polarization is zero or the momentum of annihilation gamma is zero (in fact in this case there was no o-Ps annihilation). Otherwise only special cases apply: $O_2 = 0$ for $\epsilon_i \perp \hat{k}_j$ and $O_2 = \pm 1$ for $\epsilon_i \parallel \hat{k}_j$. Therefore these discrepancies does not play a substantial role in the reported measurement. However, for the CP symmetry test the crucial observable to discuss here would be a difference between left and right side of the requested residual distribution. As visible at the right plot of Figure A the possible discrepancies between MC and data cancel out. Since the background distribution is CP symmetric for the presented operator, the expectation value of the remaining distribution (background subtracted from the data distribution) is a sensitive measure of the possible CP violation.

3. To prove both signal MC and background MC agree well with data, authors need to show all CP asymmetry-related variable plots to prove that MC and data agree well. For example, angular, and vertex distribution need to be shown with data and

MC overlapping like Extended Data Fig. 2. Extended Data Fig. 3 and Fig. 4 data need to be overlapped with signal+BG MC. Also, authors should show a plot of e_i distribution between k_2 and k_2' data overlap with signal+BG MC.

The Monte Carlo simulation used by the J-PET group is based on the Geant4 toolkit (Ref. [51]). The agreement between simulated events and experimental data for measured o-Ps annihilations and scattered gamma was already presented in previous papers by J-PET group, namely:

- Ref. [25]
Moskal, P. et al. Positronium imaging with the novel multiphoton PET scanner, Sci. Adv. 7, eabh4394 (2021).
- Ref. [43]
Dulski, K. et al. The J-PET detector – a tool for precision studies of ortho-positronium decays, Nucl. Instrum. Meth. A 1008, 165452 (2021).
- Ref. [44]
Sharma, S. et al. Estimating relationship between the Time Over Threshold and energy loss by photons in plastic scintillators used in the J-PET scanner, EJNMMI Phys 7, 39 (2020).

therefore we didn't repeat in the manuscript the discussions and comparisons presented elsewhere. However, we agree that distribution of ϵ_i is relevant. There is a most recent article devoted to scattered gamma at J-PET:

Sharma, S. et al. Efficiency determination of J-PET: first plastic scintillators-based PET scanner, EJNMMI Phys 10, 28 (2023)

where plot of Fig. 8 a is equivalent to ϵ_i distribution. For the convenience of the Reviewer we show this plot in Fig. B. We included the reference to the above-mentioned paper in the revised version of the manuscript as Ref. [45].

4. *Why there are so much up and down fluctuation in histogram theta1+theta2 histogram? If it is caused by the histogram of the binning effect, the authors need to correct it. theta1-theta2 plot needs to be added too.*

The structure of the distribution shown in Fig. 5 reflects the geometrical configuration of the scintillators in the detection system. This figure indicates that MC can describe the structure of the data very good. Left photograph of Fig. 1 shows the J-PET detector. In the foreground the metallic mounting plate is visible with three layers of the scintillator strips: two inner layers and the most outer one are used. There are two unused layers (empty slots) for additional detectors. Figure C shows the zoom of the mechanical drawing of such plate. The minimal angular distance between scintillators is equal to 1.875° . This can be seen e.g. by comparing in Figure C the placement of scintillators in the most inner and most outer layer (or by comparing the placement of scintillators in the second inner layer and the most outer layer). Fig. 2 presents the events identified as o-Ps annihilation. Such events consists of three hits forming an annihilation plane. In the analysis we require that

(a)

(b)

Figure B: Fig. 8 from Ref. [45]. **a:** Distribution of the scattering angles (θ). **b:** Distribution of the energy loss for tagged 511 keV photons. Results of the experiment and simulations are shown in blue and red, respectively. In the inset, energy deposition spectra are shown in a logarithmic scale. This figure is used to estimate the efficiency of the detection as a function of the energy deposition. In the simulations an ideal efficiency is assumed and in experiment the efficiency depends on the used electronic threshold.

this reconstructed plane has to be no more than 4 cm away from the o-Ps source position. Since the angular measurement at the XY plane (perpendicular with respect to the axis of the J-PET cylinder) is discrete (X and Y coordinates of the hits correspond to the XY center of a given strip), the discrete fluctuations have to be visible at the angular plot as well. At the discussed plot in the range $200^\circ - 230^\circ$ the 16 structures are visible. The 2D plots (left plot of Fig. 2 and Fig. 8) are used to describe the background and signal components, while for the event selections the projection on the $\theta_1 + \theta_2$ axis is used (Fig. 5). Thus, the structure seen in Fig. 5 reflects the geometrical structure of the detector and it is very well described by the

Figure C: Mechanical drawing of part of the mounting plate with holes for scintillator strips.

Monte Carlo simulations.

5. *To see the sensitivity of CP in the JPET detector, authors need to perform toy MC (which is popular in search experiments in HEP) to make sure make a 0.001 to 0.0001 level of CP violation and check whether or not authors achieve the same violation as put it in with current statistics of data.*

Similarly as in case of the operator O_1 (Eq. 1) for CPV test and operator used for CPT violation (Ref. [39]), there is no direct model which describes the violation of a given symmetry described by a peculiar operator. Therefore we decided to use a model with linear distortion of the O_2 operator described with a single parameter a , namely $Prob(O_2^{MC}) = (a \cdot \cos(\omega_{ij}) + 1) \cdot Prob(O_2)$. Result of implementation of this model is presented in Figure D. All the points are derived from the same MC statistics as used in the manuscript, therefore no error bars are presented and no statistical fluctuation are visible. It is clear, that the accuracy to be achieved on a parameter depends on the model itself. The interpretation of the a parameter in such model is out of the scope of our manuscript and the reported result should be considered as a trigger for a theoretical input. Since the conclusion about resolution on the model parameter depends on the model applied, we believe that presenting the final result in form of unbiased value of $\langle O_2 \rangle$ is justified. The plateau visible for $a < 10^{-5}$ in Figure D shows that the MC statistics used is sufficient for effects of the order of 10^{-5} . The conclusion raised here depends on the model applied. However, for the linear distortion, as used in Figure D, the methodology adopted in

the manuscript holds up to the level 10^{-4} .

Figure D: Value of the operator O_2 for MC sample only as a function of introduced CPV with an arbitrary model parameterized by a . See text for the model description.

6. How you can verify acceptance as shown in Extend Data Fig. 7 of MC in 10^{-4} level.

The unique feature of the J-PET detector, namely an access to the full range of the investigated operator, plays a crucial role here. A signal event consists of four registered hits in four different scintillator strips ($o\text{-Ps} \rightarrow 3\gamma + \gamma'$). All possible angular combinations of momenta are registered (right plot of Fig. 2 and left plot of Fig. 10) and used for the symmetry test. The most important feature of the detector is that each scintillator contribute the same way as the other scintillators to the registration of all configurations. Therefore, even if, the efficiency of a given scintillator would be not well estimated then it would modify the spectrum of O_2 operator only in amplitude but not in the shape. And only the shape is important for the test of the symmetry. In total there are 192 scintillator strips with angular distance of 1.875° at the radius of 42.5 cm and 46.75 cm. Each single strip contributes equally to the final result. For the sake of argument, full removal of a single detector (1 out of 192) would have 5×10^{-3} effect on the statistics, whereas an exaggerated misplacement of a strip by 0.1 cm, would have 0.14° effect, while the angular coverage of a single strip in the XY plane is 0.5° . So even if we would determine the efficiency and geometry of a given scintillator with the precision of 10^{-3} the effect on the total efficiency would be at the level of 10^{-6} . However, as discussed before, the total efficiency does not influence the accuracy of the determination of the expectation value for O_2 . Here the crucial factor is the relative efficiency between registration of various configurations. And since each scintillator contribute to the measurement of all configurations the determination of asymmetry in O_2 is (in the very good approximation) not affected by the inaccuracies of the of determination of the efficiency of single detector strips. However, $\theta_1 + \theta_2$ at the annihilation plane is determined with the resolution of 1.5° . Additionally, the experimentally determined resolution of the source position (from $p\text{-Po} \rightarrow 2\gamma$, see right plot at Fig. 10) is 0.5 mm in the XY

plane and 0.4 mm along the Z axis, while the resolution of distance between source position and the annihilation plane (DOP) is estimated to be 1.1 cm. Therefore we are confident of the acceptance determination at the level below 10^{-4} .

7. *Explain detail how authors got $\langle O_2 \rangle = -0.0005 \pm 0.0007$ from Fig 2 Right plot. The authors obtained 0.0007 level of statistical uncertainty with 7.7×10^5 events and more than half of background events need to be subtracted.*

The CP symmetry test with O_2 operator is based on the precision in determining the shape of the investigated distribution (right plot of Fig. 2) - an error of the mean value in the first approximation. Therefore the accuracy of this test corresponds to the standard error of the mean. The purity of the final sample is 47%. Let's simplify the signal distribution from this plot as an uniform distribution with standard deviation of 0.58. Then the standard error of the mean would be $0.58/\sqrt{0.47 \times 7.7 \times 10^5} \approx 0.00096$, which, under the assumptions made, is a good approximation of the accuracy we obtained by the method described in the "Expectation value of the correlation O_2 " section. However, the distribution is not uniform but rather enhanced at small values and therefore resulting error is smaller than 10^{-3} .

8. *Also authors need to consider MC statistics based systematic uncertainty science authors used are negligible but authors only used 3.5 and 2.4 times MC which will contribute 60% of current statistical uncertainty. Thus systematic uncertainty by MC statistics is not negligible.*

Thank you for the valuable remark. Below we will clarify the basis for our estimations which hopefully is satisfying.

The accuracy of the measurement depends on the finite statistics used (1. experimentally registered and 2. generated in the simulation) as well as the effects introduced on the way to the final result (1. during the data taking, like experimental conditions and resolutions and 2. during data analysis, like applied selection criteria). Amount of events to be considered manifests itself via statistical fluctuations of data and MC distributions and contribute to the statistical accuracy only, while the remaining effects can introduce systematical shift in the final value of the measurement. The finite number of parameters introduced during data analysis (eg. selection criteria, bin size) would contribute in a finite manner to the total systematical uncertainty. However, the number of experimental conditions which can affect the final result at any level is, in principle, "infinite", therefore the sum of such effects would be huge. The method proposed by Barlow (Ref. [52] and [53] in the revised version) allows to consider only those systematical contributions which are statistically significant. The possible systematical effect to our result are discussed in the "Estimation of systematical uncertainties" section and none of them is statistically significant. Keeping in mind the approximation from the previous point about the standard error of the mean of the signal distribution being 0.00096, we can calculate this error for the total ($\times 2.9$) MC sample: $0.58/\sqrt{2.9 \times 7.7 \times 10^5} \approx 0.00039$.

Therefore the statistics of MC sample contribute ~ 2.5 less than the signal events within experimental data sample. We consider this level as sufficient. For the simulation of events the full geometry of J-PET detector was used. This includes especially all the aluminum elements of the mechanical construction for the possible rescattered events. No influence of such events was found, however this slow down the generation process significantly.

We hope that Reviewer will agree with the publication of the result with the present precision (which is rather conservative estimation of the error). There are no CP violating effects in the Monte Carlo simulations so there will be no changes in resulting non-asymmetry if the simulations are correct. We agree that if we would increase the simulations by e.g. factor of 10 (which would be very time consuming since we simulate all effects and materials in detail), the result would become more precise. But not more than pure statistical experimental precision which is equal to 0.0006. At the same time we now started experiments with new generation of the J-PET detector with factor of about 20 higher sensitivity for the o-Positronium registration and we will work on new experiment to improve the precision by another order of magnitude.

9. Title "*Matter-antimatter symmetry tested at 10^{-4} precision*" is misleading. First, the authors tested CP violation in QED sector not matter-antimatter symmetry directly. We never call Matter-antimatter symmetry violation in the weak sector even if a large CP violation in weak sector has been observed. If authors want to put the above title, authors need to prove less than 0.001 CP violation will lead to matter-antimatter asymmetry and how much asymmetry can be predicted. Authors need to write 0.0007 (1 sigma) with only statistical uncertainty included instead of the 10^{-4} level in the abstract. Also, the authors need to make it clear that authors only test CP violations in the QED sector which is predicted to be very small, not like CP violations in the weak sector.

We are especially grateful to the Reviewer for this important remark. Matter-antimatter asymmetry in the Universe requires CP violation, baryon number violating processes and departure from thermal equilibrium in the early Universe. Our work is about the first requirement, namely testing CP, P and T symmetries. Therefore we updated the title and the new one reads: *Discrete symmetries tested at 10^{-4} precision using linear polarization of photons from positronium annihilations*. We hope it is adequate to the text and acceptable by the Reviewer.

We add the *one standard deviation* to the abstract as well. We also explicitly stated the achieved accuracy of the measurement, to keep the relation between discrepancies reported between measured hyperfine energy structure and theory.

We add a footnote at the end of section 2 "Here one is probing the discrete symmetry properties of QED. Weak interaction effects are characterized by a factor $G_F m_e^2 \approx 10^{-11}$ with G_F the Fermi constant, and would only be manifested with very much enhanced precision."

10. *In the abstract, the authors wrote "Positronium, the simplest bound state of an electron and positron, is of recent interest with discrepancies reported between measured hyperfine energy structure and theory at the level of 10^{-4} and up to 4.5 standard deviations." It is not much related to CP violation in QED that authors need to remove the above sentence in the abstract.*

In the revised version of the manuscript we made this sentence more detailed to follow suggestion of the Reviewer. We add to the abstract that: This result is "signaling a need for better understanding of the positronium system at this level."

11. *Authors need to write a 90% confidence level limit on CP violation in the QED sector.*

The accuracy quoted by us is 1σ therefore, following the Reviewer's remark, we updated the following sentence of the manuscript:

"We find a value consistent with zero at 68% confidence level, as expected from the underlying QED to factor of three more accurate than previous measurements of the CP -odd correlation,(...)"

Answers to Reviewer #2

We are grateful for recognizing importance of the motivation of our measurement. Thank you for the time devoted to read and review the manuscript. The comments you provided are important to improve the quality of this manuscript.

The paper tests P , T and CP , and thus CPT , symmetries between matter and antimatter using decays of positronium in a way independent of the measurement of the spin of the positronium, using the J-PET tomograph. They confirm preservation of CPT symmetry between matter-antimatter at the level of 10^{-4} , consistent with Quantum Electrodynamics (QED) expectations. This level of accuracy is claimed by the authors to be important, given recent anomalies observed in hyperfine structure spectroscopy measurements of positronium, leading to a 4.5 sigma discrepancies between experiment and theory (the latter being mostly non relativistic QED bound state theory).

In view of such anomalies, testing CPT symmetry independently at this level in the positronium system acquires an important meaning, and this experiment, together with the innovative approach of using the J-PET tomograph, constitutes an important platform for excluding the possibility that the aforementioned anomalies are due to a fundamental breakdown of CPT symmetry between matter and antimatter in this system.

Being a theorist, I do not have the expertise to judge the experimental details, however I can judge the importance of the motivation for this work, and, in view of the above comments, I believe the article meets the stringent criteria for being published in Nature communications.

We are happy the Reviewer #2 recognizes importance of our result.

The paper discusses in my opinion in a clear way background effects that could affect the conclusions. However, one aspect which I could not see it discussed, are the prospects for increasing the accuracy of such CPT tests beyond the 10^{-4} level. In my opinion some speculations in this direction would make the paper more complete, especially because of the importance of the subject.

In general, I consider the paper important to be published in Nature communications, provided the authors take into account my suggestion on remarking on the prospects for improved sensitivity, so as to test CPT symmetry in positronium, or other similar systems.

We truly appreciate this remark. Following the suggestion from the Reviewer we updated the manuscript with the following paragraph at the end of "Discussion" section and new Ref. [41]:

"The new result might be further improved using the methods introduced here together with upgrades in the J-PET detector. These experiments will be conducted with a modular J-PET detector having about 20 times higher sensitivity for the registration of ortho-positronium. The modular version of the J-PET system [41] with increased acceptance is currently being used for a measurement of the P , T , CP and CPT symmetries with a goal of reaching 10^{-5} accuracy."

Answers to Reviewer #3

Thank you for your insightful comments and constructive suggestions on our manuscript. We are very grateful for your time to review our work and acknowledge our methodology. We hope that our reply to your comments is sufficient to accept our work for publication.

This paper reports on a much improved test of CP in $Ps \rightarrow 3\gamma$ decays, where the polarization of one of the gammas is measured and combined with the momentum vectors of the gammas to form a CP-sensitive term that is symmetric in the case CP is conserved. Polarization of a gamma is measured by detecting a Compton scattered photon in the JPET device. The analysis is generally sound and straightforward to follow and the result represents a major advance in sensitivity; thus the paper merits publication in principle.

We appreciate that a major advance in sensitivity of our measurement is recognized.

However a small number of questions are not clear in this reviewer's mind, and would deserve explaining in more detail:

- *event selection: hit multiplicities > 3 appear to be used; with larger hit numbers, combinatorics and worsened resolutions can be expected; has the analysis also been done for $n_{hits} = 4$ events only? as a function of n_{hits} ? has an optimum n_{hits} been searched for?*

Thank you for this remark. The multiplicity distribution in Fig. 6 is presented in a logarithmic scale at vertical axis. At the beginning of the data analysis the $n_{hits} = 4$ condition was also considered but with no significant difference with respect to $n_{hits} \geq 4$. However a search for an optimum of n_{hits} was not performed, since part of the background events (also in a form of combinatorics) was required to remain in the final data sample. As explained already in the answer to the Reviewer #1 at point 2., one of the key steps of the reported measurement is the precise determination of the amount of background events in the final sample, since the shape of background and signal distributions are quite similar as visible at the right plot of Fig. 2. Therefore the normalization is done based on the distribution presented in Fig. 5. In order to do so, part of the background ($\theta_1 + \theta_2 < 200^\circ$) must be present in the final sample. Taking this into account we did not search for a specific n_{hits} , as the amount of background in the final sample is not a key issue in the reported measurement.

- *the source is not point-like, but rather somewhat extended (5mm radius, several mm in length, 1-3 mm thickness); while I can't think of any asymmetry that could result from this extensive volume (in which gammas can scatter), I wonder how this rather large material budget can affect the resulting distributions. One concern might be that 2gamma decays (accompanied by background hits) could more easily enter the 4 hit candidate sample as scattering would reduce the 180 degree opening angle.*

For the convenience of the Reviewer the details of the annihilation chamber are

presented in Figure E. The chamber is built from PA6 plastic with the thickness of

Figure E: Photo (top) and detailed scheme of the annihilation chamber.

chamber walls in the active range of the detector is less than 1 mm (Ref. [31]). The attenuation of photons from o-Ps annihilation due to material budget is estimated to 1% (as mentioned in "Experimental setup" section) for all orientations of annihilation plane that can be registered by the J-PET setup. The Reviewer is right about possible p-Ps $\rightarrow 2\gamma$ contribution to the final data sample due to scattering. Such contribution is visible in Fig. 5 and 8 for $\theta_1 + \theta_2$ close to 180° . However, it is mostly from the detector scatterings rather than source/chamber.

- *Fig. 2 left shows the experimental data, while the MC ingredients are shown in the supplementary material; Fig. 2 right shows the O_2 variable for the same data sets. How is the normalization (pg. 16) carried out? I am concerned that the simulation/fit of signal and backgrounds very poorly reproduces the experimental distribution in the small O_2 region, and even more so that the experimental distribution appears to have an asymmetry with respect to the (symmetric) MC in the second (and to a much smaller extent, the third) bin (0.1-0.2, resp. 0.2-0.3). Given the invisible error bars on the experimental points (presumably lying within the circles), the discrepancy is highly significant...*

Thank you for the remark. We have considered various types of the background. Contribution of the events originating from the cosmic radiation is negligible as was established in a dedicated measurement without the radioactive source. Therefore

only contributions from $p\text{-Ps} \rightarrow 2\gamma$ and independently from $o\text{-Ps} \rightarrow 3\gamma$ were considered in the simulations. The possible accidental coincidences are negligible, due to small activity of up to 5 MBq used in the measurement. The relative amount of signal and background contributions is calculated on the basis of the distribution shown in Fig. 5. A fit with two free parameters is performed. Each parameter being a scaling factor of a given contribution. The best parameters are obtained for the smallest χ^2 value calculated from a sum over all bins (for data points and a sum of simulated contributions).

Analogously as discussed also in the answer to the Reviewer #1 at point 2 and Figure A. For the convenience of the Reviewer #3 we add here Figure A again as Figure F. One of the novelties of our measurement is an access to the full phase

Figure F: **Left:** Plot of percentage residual for O_2 operator defined as $\text{abs}(\text{data-MC})/\text{data}$. **Right:** Difference between bins of O_2 operator (from the left plot) of negative and positive values, respectively. The black line is to guide the eye only.

space of the operator O_2 . Therefore the performed test of the CP symmetry is based on the full spectrum of the operator values and, as such, the expectation value is equivalent to the mean value of distribution of O_2 . The achieved sensitivity is due to acquired statistics as well as the total range of the operator values. Two crucial points here are: estimation of the distribution of acquired signal events (by subtraction of MC background events from the data distribution) in the function of O_2 and the overall agreement between MC and data distribution in the function of O_2 . The key factor here is that in order to avoid vicious circle the Monte Carlo parameters, like background and signal normalization factors, were estimated at independent distribution (Fig. 5). Therefore, this distribution presents better MC to data agreement with respect to Fig. 2. These normalization factors were later on used for MC distributions in Fig. 2. The biggest discrepancies between MC and data (Figure F) are for the side and central regions of O_2 values. It is important to stress here, that due to the nature of the O_2 operator itself the number of events tends to zero for $O_2 = 0$. Following Eqs. 2-4 from the manuscript, $O_2 = 0$ when the scattering occurred under 0° angle (no scattering) and therefore the determined polarization is zero or the momentum of annihilation gamma is zero (in fact in this

case there was no o-Ps annihilation). Otherwise only special cases apply: $O_2 = 0$ for $\epsilon_i \perp \hat{k}_j$ and $O_2 = \pm 1$ for $\epsilon_i \parallel \hat{k}_j$. Therefore these discrepancies does not play a substantial role in the reported measurement. However, for the CP symmetry test the crucial observable to discuss here would be a difference between left and right side of the requested residual distribution. As visible at the right plot of Figure A the possible discrepancies between MC and data cancel out. Since the background distribution is CP symmetric for the presented operator, the expectation value of the remaining distribution (background subtracted from the data distribution) is a sensitive measure of the possible CP violation.

- *a related figure regards Fig. 2 left: what does the residual 2-d distribution look like of the MC are scaled according to Fig. 2 right and subtracted from the experimental data of Fig 2 left?*

An equivalent distribution is presented in Figure G.

Figure G: Distribution of $\theta_2 - \theta_1$ vs $\theta_1 + \theta_2$ for the $|data - MC|/data$. The statistical fluctuations are randomly spread in the whole region of interest. The signal is localized mostly in the $\theta_1 + \theta_2 > 200^\circ$ region.

- *is the O_2 distribution in Fig. 2 right corrected for detection efficiency (suppl. fig 7)? What causes the enhanced/suppressed structures (1st, 2nd, 3rd bins) in both of these distributions?*

We use the Monte Carlo simulations for understanding the detector performance and for the efficiency corrections. However, it is important to stress that the raw, not corrected result is also showing no CP violation and is consistent with the final corrected result. Therefore the final result does not depend on the corrections. The detector system was especially designed to be symmetric such that in principle it does not introduces artificial asymmetries. This is because each out of 192 scintillator strips contributes to the registration of all O_2 values ("configurations"). Moreover, all configurations of the o-Ps $\rightarrow 3\gamma + \gamma_{scatter}$ are measured simultaneously

without a need to change detector configurations. Therefore, the corrections simulated by the Monte Carlo method will not affect the symmetry or asymmetry of O_2 operator observed in the experimental data.

There are two usual approaches for presenting experimental results. Either one should not modify the experimental data at any cost and provide an acceptance (efficiency) distribution, or one should obtain a distribution as close to physical phenomenon as possible (meaning applying acceptance correction to the experimental data). Here we follow the first approach, we provided the efficiency distribution (left plot of Fig. 10) and therefore distribution presented at the right plot of Fig. 2 is not corrected for acceptance and efficiency. However, the final result is calculated from the corrected distribution.

The acceptance increases for $O_2 \rightarrow 0$, since most of scatterings occur under relatively small angle for plastic scintillators. However, at the same time for $O_2 \rightarrow 0$ the efficiency goes to 0, since a small scattering angle is equivalent to a small (undetectable) energy deposition (Ref. [21] and [22]). These two asymptotic effects combined in just three bins are resulting in observed distortion in the central region of O_2 distribution.

Additional minor questions are:

- *pg. 9 coplanarity of photons is an important selection variable, but the experimental distribution (and the backgrounds) is not provided...perhaps something to add to the supplementary material*

In the reported measurement the coplanarity of three photons is considered as a distance between annihilation plane (defined by three hits of annihilation photons) and the source position, called as distance of the plane (DOP) variable. Experimental and signal distributions of DOP are presented at the bottom left plot of Fig. 7.

- *page 9, bottom: Do you mean that the Monte Carlo background is subtracted from the experimental distribution? The sentence reads ambiguously and would better be inverted.*

We are grateful for this remark. We updated the text accordingly: "For the distribution of O_2 operator the background expected on the grounds of performed Monte Carlo simulations is subtracted from the experimental distribution."

- *pg. 10: what do you mean by "complex" in "Positronium is the simplest complex bound state"? Isn't it the simplest bound state, together with hydrogen?*

We are grateful for this remark. We decided to remove this sentence in the revised manuscript to avoid any confusion.

- *pg. 14, second bullet: what is the energy of the ^{22}Ne deexcitation photon?*

The energy of deexcitation photon from $^{22}\text{Ne}^*$ is 1275 keV. The registered TOT=17 ns corresponds to the center of the Compton edge of 511 keV photons, therefore hits with TOT > 17 ns are not considered in the analysis.

- *the concept of ETS (and that of DOP) is not explained clearly (pg. 14 and extended data fig. 4) - presumably, the time of flight is calculated from the position of the detected hit for each photon, and the time of emission of the three photons reconstructed for a source assumed to lie at (0,0,0)? An equation or a sketch might help.*

We are grateful for this remark. We updated the text accordingly:

”The identification of hits from o-Ps $\rightarrow 3\gamma$ decay was performed as follows:

- the emission time was calculated for each hit as a difference between the registered time (hit time) and a travel time (ratio of distance between source and hit position and speed of light); the emission time spread (ETS) was calculated as a difference between last and first emission time of three candidates; this ETS must be less than or equal to 1.4 ns to ensure that hits originate from the same o-Ps decay (the top right panel of Fig. 7);
 - for a source position of (s_x, s_y, s_z) a distance between annihilation plane (spanned by the annihilation photons’ momenta and defined as $Ax + By + Cz + D = 0$) and the source was calculated as $\text{DOP} = |A \cdot s_x + B \cdot s_y + C \cdot s_z + D| \cdot (A^2 + B^2 + C^2)^{-\frac{1}{2}}$; the DOP constructed with three candidate hits must be less than or equal to 4 cm to reject hits from multiple scatterings (see the bottom left panel of Fig. 7);
 - at the decay plane the sum of two smallest angles between photon momentum vectors from o-Ps $\rightarrow 3\gamma$ decay must be greater than or equal to 190° (Fig. 2, 5 and 8) to reject main contribution from p-Ps $\rightarrow 2\gamma$ events with multiple scattered photons;”
 - for events with more than three hits a combination with the smallest $(\text{ETS})^2 + (\text{DOP})^2$ value was selected;”
- *have you developed a better proxy for photon energy via the use of multiple threshold TOT_s (suppl. fig. 1)? How does the calculation of the photon energies from the overall geometry of their emission directions and the assumption of $\text{Sum}_E=2m_e$ compare with such an optimized proxy? What is the resolution of E_γ when the approach of pg. 9 is used? Does the resolution of 2-gamma event TOT’s match that of the background sample in the 4-gamma distribution of suppl. data fig. 4 top left?*

The method suggested by the Reviewer was already considered by our group as described by Eq. 7 in the Ref. [37]. The achieved energy resolution is 4 keV (sigma), however it can not be used for o-Ps identification as the $\sum_{n=1}^3 E_i = 2m_e$ condition is embedded in the method itself. The interaction of gamma with plastic scintillator goes via Compton scattering, therefore a deposited energy is determined instead of

energy of gamma itself. The measurement of Time-Over-Threshold is equivalent to the measurement of deposited energy at J-PET. The estimated equivalent of the deposited energy resolution for the reported measurement is 14 keV (sigma) as mentioned in the "Experimental setup" section, while the general energy resolution for J-PET is given by Eq. 4 in the Ref. [37].

We have also determined the relation between the energy deposition and the measured TOT. This was a dedicated investigation published in Ref. [44] and [45]. In general we find that the relation between TOT and energy loss is not linear but it is well defined and we have determined phenomenological formula which relates these two quantities. Detailed discussion concerning TOT and E_{dep} relation as well as MC to data comparison for scattered events can be found in Ref. [44] (Fig. 11) and [45] (Fig. 8).

For the convenience of the Reviewer we show these plots in Fig. H and Fig. I, respectively. We included the reference to the above-mentioned paper in the revised version of the manuscript as Ref. [45].

In addition to the above, on a large number of occasions, English awkwardness (missing particle, formulation) is apparent in the text; given their number, listing all of them would be excessive, but in many cases "the" or "a" is missing, or sentences would benefit from being rewritten by a native speaker.

Multiple corrections of the language were applied by a native speaker in the revised version of the manuscript.

(a)

(b)

Figure H: Fig. 11 from Ref. [44]. **a:** 2-D spectrum of TOT versus energy deposition. **b:** TOT vs energy deposition. Black rectangles correspond to the experimental data. The statistical errors in measuring the values are smaller than the size of symbols due to the large number of entries for the fitted TOT distributions. The red dashed line indicates result of the fit of the function: $TOT = A0 + A1 * \ln(E_{dep} + A2) + A3 * (\ln(E_{dep} + A2))^2$, with $A0 = -2322$ ns, $A1 = 632.1$ ns/keV, $A2 = 590.2$ keV, and $A3 = -42.29$ ns/(keV)². Green dotted line shows the result of another fitting function $TOT = A0 - A1 * A2^{E_{dep}}$ with three parameters where $A0 = 42.96$ ns, $A1 = 53.43$ ns, and $A2 = 0.997$ keV⁻¹. Blue dotted-dashed line represents the model predictions (the “Theoretical model” section of Ref. [44]) for the total TOT values at four fixed thresholds for the time distribution spectra. In framework of J-PET, the light signals are collected on both sides of plastic scintillator as a measure of energy deposition, so in calculating the TOT model, we used twice the value of TOT sum and the value of free parameter $N0$ is 1.3.

(a)

(b)

Figure I: Fig. 8 from Ref. [45]. **a**: Distribution of the scattering angles (θ). **b**: Distribution of the energy loss for tagged 511 keV photons. Results of the experiment and simulations are shown in blue and red, respectively. In the inset, energy deposition spectra are shown in a logarithmic scale. This figure is used to estimate the efficiency of the detection as a function of the energy deposition. In the simulations an ideal efficiency is assumed and in experiment the efficiency depends on the used electronic threshold.

Other changes applied to the revised version

We adopted numbering of figures to the scheme of Nature Communications, therefore the order of figures is changed. We also discovered an embarrassing typo in the originally submitted manuscript. The graphical representation of the final result in Fig. 3 is correct. However there was an unintentionally placed minus sign in the Eq. 5. The revised version of the manuscript is corrected.

REVIEWER COMMENTS

Reviewer #1 (Remarks to the Author):

The authors put a lot of effort, into either revising the manuscript or answering all my comments that I recommend this manuscript for publication.

However, I recommend that authors either find a method to rely less on MC simulation for the improvement of limits, or authors need to find a way to estimate systematic error accurately for future publication.

Reviewer #3 (Remarks to the Author):

I thank the authors for several clarifications, and I consider that most of my concerns have been satisfactorily addressed. One point however remains where the authors' explanations have not fully convinced me, or where I remain confused by them, and this concerns figure 2.

In my question, I had pointed out the discrepancies between the simulation and the data in the low O₂ region, as well as an asymmetry. Specifically, my question concerned:

- the quality of the simulation with a significant (15 sigma!) discrepancy in the two central bins;
- the possible origin of the experimental asymmetry between the [-0.2,-0.1] and the [0.1, 0.2] bins.
- to which I now add a question on the systematic underestimate in the [-0.3,0.3] and systematic overestimate in the [-1.0,-0.3] and [0.3,1.0] regions.

To the second point, an answer could have been that the two values are statistically compatible with each other, in spite of the apparent difference in figure 2 right. I can somewhat convince myself of that via Figure F left.

I understand that the normalization comes from figure 5, and imposes the relative ratio of background and signal distributions in figure 2. The strong discrepancy between the two however tells me that either the signal or the background(or both) simulations are

systematically (slightly) wrong.

The reply by the authors is that the CP sensitivity comes from the overall distribution, underlined by their statement in their reply that "possible discrepancies between MC and data cancel out" (Figure F right). This plot is stated to stem from the differences in figure F left between bins of negative and positive values (caption of figure F). I do not understand how they obtain this plot. If I look at P- - P+ bin by bin from figure F left, I get (approximate values):

$$[-0.1,0.0] - [0.0,0.1]: 0.075 - 0.08 \sim -0.005 \pm 0.005$$

$$[-0.2,-0.1] - [0.1,0.2]: 0.03 - 0.045 \sim -0.015 \pm 0.005$$

...

$$[-0.9,-0.8] - [0.8,0.9]: 0.035 - 0.05 \sim -0.015 \pm 0.005$$

$$[-1.0,-0.9] - [0.9,1.0]: 0.155 - 0.145 \sim +0.01 \pm 0.01$$

This does not correspond to figure F right: the first bin is positive there, the next negative, the penultimate and last both negative, and the non-zero value of the second bin is within the error of zero, contrary to the above values. I have thus either misunderstood how figure F right is constructed, or the provided plot is incorrect (I don't think that the signs of the differences would change if I were to use the raw counts of figure 2 right). In both cases, I have to withhold judgement on whether indeed, overall, the discrepancies cancel out (a more precise formulation would be that there is no evidence of a trend in figure F right that could indicate a systematic difference between positive and negative O₂ values).

The 15 sigma discrepancy between data and simulation for small O₂ values may indeed not be critical, but it does raise a worry about the detailed understanding of the apparatus. At this point, I believe the authors that they have not been able to think of further backgrounds, but a residual concern remains that something significant is missing. Specifically, it looks like the shapes of the simulated signal and background O₂ distributions may be slightly incorrect (too broad; as a guess, if the signal distribution were slightly 'smeared' in the center and 'pulled in' for larger O₂ values, the agreement would be better - is thus the signal simulation somewhat off?). In light of the fact that indeed, the O₂ distribution in figure 2 is left-right symmetric, this might be nit-picking, but we are talking

about searches for minute differences. Perhaps a simple statement admitting that the origin of the discrepancy is not understood, but only plays a marginal (perhaps to be quantified) role on the CP limit could be added.

Below we answer point by point to the Reviewers' remarks. The modifications of the manuscript itself are marked with red. The comments of the Reviewers are quoted in italics for convenience.

Answers to Reviewer #1

The authors put a lot of effort, into either revising the manuscript or answering all my comments that I recommend this manuscript for publication. However, I recommend that authors either find a method to rely less on MC simulation for the improvement of limits, or authors need to find a way to estimate systematic error accurately for future publication.

We are grateful for all the suggestions from the Reviewer #1 and for acceptance of the revised version of our manuscript. Our result might be further improved using a modular version of J-PET detector. Since this version of the detector has about 20 times higher sensitivity for the registration of ortho-positronium, there will be an opportunity to determine a data based correction for the acceptance and the efficiency.

Answers to Reviewer #3

I thank the authors for several clarifications, and I consider that most of my concerns have been satisfactorily addressed.

We appreciate the work dedicated to follow our response and the revised manuscript.

One point however remains where the authors' explanations have not fully convinced me, or where I remain confused by them, and this concerns figure 2. In my question, I had pointed out the discrepancies between the simulation and the data in the low O_2 region, as well as an asymmetry. Specifically, my question concerned:

- *the quality of the simulation with a significant (15 sigma!) discrepancy in the two central bins;*

- *the possible origin of the experimental asymmetry between the $[-0.2,-0.1]$ and the $[0.1, 0.2]$ bins.*

- *to which I now add a question on the systematic underestimate in the $[-0.3,0.3]$ and systematic overestimate in the $[-1.0,-0.3]$ and $[0.3,1.0]$ regions.*

To the second point, an answer could have been that the two values are statistically compatible with each other, in spite of the apparent difference in figure 2 right. I can somewhat convince myself of that via Figure F left.

I understand that the normalization comes from figure 5, and imposes the relative ratio of background and signal distributions in figure 2. The strong discrepancy between the two however tells me that either the signal or the background(or both) simulations are systematically (slightly) wrong.

The reply by the authors is that the CP sensitivity comes from the overall distribution,

underlined by their statement in their reply that "possible discrepancies between MC and data cancel out" (Figure F right). This plot is stated to stem from the differences in figure F left between bins of negative and positive values (caption of figure F). I do not understand how they obtain this plot. If I look at P- - P+ bin by bin from figure F left, I get (approximate values):

$$[-0.1,0.0] - [0.0,0.1]: 0.075 - 0.08 \sim -0.005 \pm 0.005$$

$$[-0.2,-0.1] - [0.1,0.2]: 0.03 - 0.045 \sim -0.015 \pm 0.005$$

...

$$[-0.9,-0.8] - [0.8,0.9]: 0.035 - 0.05 \sim -0.015 \pm 0.005$$

$$[-1.0,-0.9] - [0.9,1.0]: 0.155 - 0.145 \sim +0.01 \pm 0.01$$

This does not correspond to figure F right: the first bin is positive there, the next negative, the penultimate and last both negative, and the non-zero value of the second bin is within the error of zero, contrary to the above values. I have thus either misunderstood how figure F right is constructed, or the provided plot is incorrect (I don't think that the signs of the differences would change if I were to use the raw counts of figure 2 right). In both cases, I have to withhold judgement on whether indeed, overall, the discrepancies cancel out (a more precise formulation would be that there is no evidence of a trend in figure F right that could indicate a systematic difference between positive and negative O_2 values).

The 15 sigma discrepancy between data and simulation for small O_2 values may indeed not be critical, but it does raise a worry about the detailed understanding of the apparatus. At this point, I believe the authors that they have not been able to think of further backgrounds, but a residual concern remains that something significant is missing. Specifically, it looks like the shapes of the simulated signal and background O_2 distributions may be slightly incorrect (too broad; as a guess, if the signal distribution were slightly 'smeared' in the center and 'pulled in' for larger O_2 values, the agreement would be better - is thus the signal simulation somewhat off?). In light of the fact that indeed, the O_2 distribution in figure 2 is left-right symmetric, this might be nit-picking, but we are talking about searches for minute differences. Perhaps a simple statement admitting that the origin of the discrepancy is not understood, but only plays a marginal (perhaps to be quantified) role on the CP limit could be added.

We believe all three points raised by the Reviewer are strongly correlated, therefore we discussed them together below:

1. the quality of the simulation with a significant (15 sigma!) discrepancy in the two central bins.
2. the possible origin of the experimental asymmetry between the [-0.2,-0.1] and the [0.1, 0.2] bins.
3. the systematic underestimate in the [-0.3,0.3] and systematic overestimate in the [-1.0,-0.3] and [0.3,1.0] regions.

The accuracy of the MC simulation with respect to the data is presented on the left plot of Fig. A. For the convenience of the Reviewer it is the same plot as in the previous response. However, as pointed out by the Reviewer, the corresponding plot of a difference

between bins of O_2 operator of negative and positive values was incorrect. We are grateful for checking carefully the figure and we appologize for the confusion due to this mistake. In the plot attached to the previous response the order of the points was reversed. The right plot of Fig. A is correct and the approximate estimations performed by the Reviewer are generally appropriate, e.g. for the penultimate and last point it should be:

$[-0.9,-0.8] - [0.8,0.9]: 0.037 - 0.053 \sim -0.016 \pm 0.015$
 $[-1.0,-0.9] - [0.9,1.0]: 0.155 - 0.135 \sim +0.02 \pm 0.02$

Figure A: **Left:** Plot of percentage residual for O_2 operator defined as $\text{abs}(\text{data-MC})/\text{data}$. (The same plot as in the previous response to the Reviewer.) **Right:** Difference between bins of O_2 operator (from the left plot) of negative and positive values, respectively. The black line is to guide the eye only. (The points presented here are swap in a mirrored left-right way with respect to the corresponding plot in the previous response.)

Figure B: Monte Carlo simulation derived distribution of geometrical acceptance of the detector (left) and efficiency including all selection criteria applied for the described analysis for signal events (right).

All points but one show no asymmetry within 1σ on the right plot of Fig. A.

Fig. 10 in the manuscript is a combined distribution of geometrical acceptance of the detector (left plot on Fig. B) and efficiency including all selection criteria applied for the described analysis for signal events (right plot on Fig. B).

The rapid and opposite change on the acceptance and efficiency in the central region of O_2 operator is clearly visible. The interplay between the acceptance and the efficiency manifests itself in the *fluctuation* of $[-0.2, -0.1]$ and $[0.1, 0.2]$ bins on Fig. 10 in the manuscript. (Please keep in mind that the right plot on Fig. 2 of the manuscript is not corrected for the efficiency and acceptance.) Additionally, the true efficiency goes to zero for $|O_2| \rightarrow 0$. This strong change may increase any possible statistical fluctuation in that region, which may be responsible for the small asymmetry between the $[-0.2, -0.1]$ and the $[0.1, 0.2]$ bins (as questioned also by the Reviewer in the second point). Nevertheless, we cannot find any detector based explanation for this experimental asymmetry. On the other hand, the same behavior of efficiency distribution may cause a problem for MC simulation with finite precision (and discrete bin size at O_2 distribution), which can describe a discrepancy between MC and data for the two most central bins (first point raised by the Reviewer). As this discrepancy is visible only for the central bins and, in the worst case scenario it would correspond for the underestimated background contribution, the overall effect would be less than 10^{-5} . Therefore, following the suggestion from the Reviewer, we included the following sentence in the caption of Fig. 2:

”The discrepancy between simulated distribution and data points for the two central bins may be explained by the rapid change of efficiency distribution in that region, but this effect is negligible comparing to the achieved accuracy of the final result.”

We agree with the observation noted by the Reviewer in the third point. We agree also that inclusion of additional smearing to signal (and/or background) events on the right plot of Fig. 2 would improve overall agreement between MC and data, namely reduced underestimation in the $[-0.3, 0.3]$ region and overestimation elsewhere. Additionally it would reduce discrepancy for the two most central bins discussed for the first point raised by the Reviewer. However, all the effect due to experimental resolutions were confirmed on Fig. 5 where the normalization of MC to data was also performed. Therefore any additional smearing to improve MC to data agreement would have to be applied to the right plot of Fig. 2 only. This would require to perform the normalization directly with distributions from the right plot of Fig. 2 and therefore entering a vicious circle: normalization and final result determination on the same plot. Therefore we decided to stick with maybe not perfect agreement between MC and data on the right plot of Fig. 2, but having an independent method for determination of MC parameters. We hope the Reviewer would accept this approach.

REVIEWERS' COMMENTS

Reviewer #3 (Remarks to the Author):

I thank the authors for their detailed and honest answer and for having clarified those elements that had led to my earlier concerns. I consider that their reply has satisfactorily addressed those concerns, and am happy to recommend publication of their paper.